# Prominin-1 controls stem cell activation by orchestrating ciliary dynamics

Donald Singer[1,†], Kristina Thamm[2,†], Heng Zhuang[1,3,†], Jana Karbanová[2], Yan Gao[1,4], Jemma Victoria Walker[1], Heng Jin[5,6], Xiangnan Wu[7], Clarissa R Coveney[8], Pauline Marangoni[7], Dongmei Lu[9], Portia Rebecca Clare Grayson[1], Tulay Gulsen[1,‡], Karen J Liu[10], Stefano Ardu[11], Angus KT Wann[8], Shouqing Luo[12] ID, Alexander C Zambon[13], Anton M Jetten[14] ID, Christopher Tredwin[1], Ophir D Klein[7,15] ID, Massimo Attanasio[5], Peter Carmeliet[16,17] ID, Wieland B Huttner[18] ID, Denis Corbeil[2,*] ID & Bing Hu[1,**] ID

## Abstract

Proper temporal and spatial activation of stem cells relies on highly coordinated cell signaling. The primary cilium is the sensory organelle that is responsible for transmitting extracellular signals into a cell. Primary cilium size, architecture, and assembly–disassembly dynamics are under rigid cell cycle-dependent control. Using mouse incisor tooth epithelia as a model, we show that ciliary dynamics in stem cells require the proper functions of a cholesterol-binding membrane glycoprotein, Prominin-1 (Prom1/CD133), which controls sequential recruitment of ciliary membrane components, histone deacetylase, and transcription factors. Nuclear translocation of Prom1 and these molecules is particularly evident in transit amplifying cells, the immediate derivatives of stem cells. The absence of Prom1 impairs ciliary dynamics and abolishes the growth stimulation effects of sonic hedgehog (SHH) treatment, resulting in the disruption of stem cell quiescence maintenance and activation. We propose that Prom1 is a key regulator ensuring appropriate response of stem cells to extracellular signals, with important implications for development, regeneration, and diseases.

Keywords CD133; cilia; sonic hedgehog; stem cells; tooth
Subject Categories Cell Adhesion, Polarity & Cytoskeleton; Development & Differentiation; Stem Cells
The EMBO Journal (2019) 38: e99845

## Introduction

Tissue homeostasis depends on proper stem cell maintenance and activation to guarantee that balanced cell lineages can be produced upon necessity (Li & Clevers, 2010; Cheung & Rando, 2013). Stem cell hypoactivation can have severe consequence such as regeneration failure (Chen *et al*, 2012), while stem cell hyperactivation or aging can either exhaust the stem cell pool (Flach *et al*, 2014) or

1   Peninsula Dental School, University of Plymouth, Plymouth, UK
2   Tissue Engineering Laboratories, Biotechnology Center and Center for Molecular and Cellular Bioengineering, Technische Universität Dresden, Dresden, Germany
3   Department of Cariology, Endodontology and Operative Dentistry, Peking University School and Hospital of Stomatology, Beijing, China
4   Department of Orthodontics, School of Stomatology, Capital Medical University, Beijing, China
5   Department of Internal Medicine, University of Iowa, Iowa City, IA, USA
6   Department of Emergency Medicine, Tianjin Medical University General Hospital, Tianjin, China
7   Program in Craniofacial Biology and Department of Orofacial Sciences, University of California, San Francisco, CA, USA
8   Arthritis Research UK Centre for Osteoarthritis Pathogenesis, Kennedy Institute, Nuffield Department for Orthopaedics, Rheumatology, and Musculoskeletal Sciences, University of Oxford, Oxford, UK
9   Department of Internal Medicine, University of Texas Southwestern Medical Center, Dallas, TX, USA
10  Centre for Craniofacial and Regenerative Biology, King's College London, London, UK
11  Division of Cariology & Endodontology, Dental School, University of Geneva, Geneva, Switzerland
12  Peninsula Medical School, University of Plymouth, Plymouth, UK
13  Biopharmaceutical Sciences, Keck Graduate Institute, Claremont, CA, USA
14  Cell Biology Section, Division of Intramural Research, National Institute of Environmental Health Sciences, National Institutes of Health, Research Triangle Park, NC, USA
15  Department of Pediatrics and Institute for Human Genetics, University of California, San Francisco, CA, USA
16  Department of Oncology, Laboratory of Angiogenesis and Vascular Metabolism, KU Leuven, Leuven, Belgium
17  VIB-KU Leuven Center for Cancer Biology, Leuven, Belgium
18  Max Planck Institute of Molecular Cell Biology and Genetics, Dresden, Germany
    *Corresponding author. Tel: +49 351 463 40118; E-mail: denis.corbeil@tu-dresden.de
    **Corresponding author. Tel: +44 1752 437 804; E-mail: bing.hu@plymouth.ac.uk
    †These authors contributed equally to this work
    ‡Present address: University College London Cancer Institute, London, UK

initiate oncogenesis (Reya *et al*, 2001; Plaks *et al*, 2015). At quiescence, stem cells produce long primary cilia that are responsible for sensing and transmitting signals into the cells (Izawa *et al*, 2015; Sanchez & Dynlacht, 2016). Stem cell activation is accompanied by dynamic primary cilium size, architecture, and assembly–disassembly modifications (Izawa *et al*, 2015; Sanchez & Dynlacht, 2016) and release of factors from primary cilium to cytoplasm to trigger downstream cascades that are responsible for cell lineage differentiation (Berbari *et al*, 2009; Goetz & Anderson, 2010). Most evidence for primary cilium participation in stem cell activation and renewal has been based on cultured cells. Historically it has been difficult to investigate stem cell ciliary dynamics *in vivo* since very few model systems possess enriched stem cells and transit amplifying cells at the same time and location. Therefore, the molecular mechanisms of orchestrating primary cilium assembly and its impact on stem cell fate determination have not been fully understood yet in tissue/organ level. Here, we use continuously growing mouse incisor as a model where epithelial stem cells represent a large proportion of cells at the distal end of the tooth epithelium named cervical loop (CL) (Jussila & Thesleff, 2012; Biehs *et al*, 2013) that provides a dynamic system in which we are able to profile the functional associations between primary cilia and stem cells.

To gain insight into the precise molecular cues orchestrating primary cilium elements recruitment, we focus on one key stem cell marker called Prominin-1 (Prom1, also known as CD133), which is specifically associated with plasmalemmal protrusions (e.g., microvilli and primary cilium) (Corbeil *et al*, 2001). Prom1 is widely used in identifying stem cells in various somatic tissues and initiating cancer stem cells (Grosse-Gehling *et al*, 2013). In cancer, numerous lines of evidences suggest Prom1[high] cancer stem cells are resistant to radio- and chemo-therapies and Prom1 neutralizing antibodies or targeted toxin have anti-tumor activity through unclear mechanism (Damek-Poprawa *et al*, 2011; Waldron *et al*, 2011). *PROM1* mutations cause various human retinal disorders by disrupting the cilium-derived photoreceptor outer segment (Fargeas *et al*, 2015), a phenomenon phenocopied in *Prom1* null mice (Zacchigna *et al*, 2009). In photoreceptors, Prom1 interacts with protocadherin 21 (Yang *et al*, 2008), while other lipid and protein interactors were identified in various biological systems (Röper *et al*, 2000; Karbanová *et al*, 2008; Mak *et al*, 2012) making this pentaspan membrane glycoprotein, or other members of prominin family (Fargeas *et al*, 2003), a potential mechanoprotein in the organization of plasma membrane protrusion, and hence influencing the signaling pathways associated with them. Furthermore, extracellular membrane vesicles found in various body fluids were identified and immunoisolated based on the presence of Prom1 (Marzesco *et al*, 2005). They are released either by the membrane budding from plasmalemmal protrusions (Dubreuil *et al*, 2007) or upon the fusion of multivesicular bodies with plasma membrane and discharged as exosomes (Bauer *et al*, 2011). Prom1[+] vesicles are upregulated in cerebrospinal fluid of patients with gliomas or degenerative neurological disease (Huttner *et al*, 2008; Bobinger *et al*, 2017).

If and how Prom1 can be used as a molecular biological tool in regulating stem cells for regenerative medicine and anti-cancer therapies are still uncertain. We now show that Prom1 has essential roles in recruiting the primary cilium molecular compartments and in nuclear translocation that are critical for stem cell activation and homeostasis.

# Results

## Mouse incisor epithelial stem cells have a dynamic primary cilium biogenesis

The mouse incisor cervical loop epithelium (CLE) has a significant transit amplifying cell pool that is adjacent, but molecularly distinct from the stem cells (Appendix Fig S1A). The two cell populations could be distinguished for instance, at postnatal day 7 (P7) lower incisor CLE by immunofluorescence (IF) using markers Sox2 and Bmi1 for stem cells, and Ki67 for transit amplifying cells (Fig 1A and Appendix Fig S1B) or upon laser capture microdissection followed by real-time RT–PCR profiling using markers such as Sox2, Bmi1, CDKn1c, CDKn2b, and p16 for stem cells, and CDK5r1, c-Myc, and Sonic Hedgehog (SHH) for transit amplifying cells (Fig 1B and C). Neighboring to the transit amplifying cells, we identified also a stem cell zone harboring scattered clusters of Ki67-positive cells that represented activated and self-renewing stem cells (Fig 1A). Hence, the CLE system enabled us to monitor temporal and spatial changes in ciliary dynamics during stem cell activation and self-renewal. To validate the primary cilium profiles in the CLE, we performed immunostaining on the primary cilium axoneme (or the core) using anti-acetylated α-tubulin (AcTub) antibodies followed by three-dimensional (3D) measurements to determine whether cilium size and integrity were linked to the stem cell status *in vivo* (Appendix Fig S1C). Consistent with the conventional cilium dynamic and cell cycle linkage concept, we confirmed that the CLE-associated stem cells (CLESCs) had longer and larger primary cilia and possessed a higher number of cells retaining them comparing to the transit amplifying cells (Fig 1D–G and Appendix Figs S1D and E).

Primary cilium axoneme assembly–disassembly and cilium membrane formation are closely coordinated cellular events (Sanchez & Dynlacht, 2016). Intraflagellar Transport 88 (IFT88) is one of the key intrinsic cilium components associating with centrosomes and moving along ciliary microtubules during cilium biogenesis to transport molecules such as the SHH pathway members (Pazour *et al*, 2002). Deletion of *IFT88* in neural crest-derived cells or mesenchymal cells causes severe craniofacial deformities (Tian *et al*, 2017). To investigate how primary cilium functions in the CLESCs, we therefore temporally deleted *IFT88* by crossing *IFT88[flox/flox]* mice (Haycraft *et al*, 2007) with tamoxifen-inducible *Rosa26[CreER]* transgenic mice (Badea *et al*, 2003). After the Cre recombinase was activated for 10 days starting from postnatal day 21 (P21) (Fig 1H), we first analyzed the regions where Cre was expressed. Interestingly in the CLE, the Cre expression was highly focused on the stem cells, while in the mesenchyme, the expression is more homogenous (Appendix Fig S1F). Next, we observed a significant reduction of the total stem cell zone volume using 3D reconstruction based on Sox2-positive immunolabeling (Fig 1H–J). The amount of proliferating Ki67[+] cells was considerably reduced (Fig 1H). Hence, disrupting cilium dynamics in the stem cells impeded stem cell renewal and activation.

Besides general intrinsic cilium molecules such as IFT88, stem cells need to recruit temporary specific molecular players regulating signaling pathway(s) associated with primary cilia to respond to external stimulation. The SHH pathway is one key stem cell-related signaling in the incisor CLESCs (Seidel *et al*, 2010; Zhao *et al*, 2014). In response to SHH signal, a cell needs to recruit or express

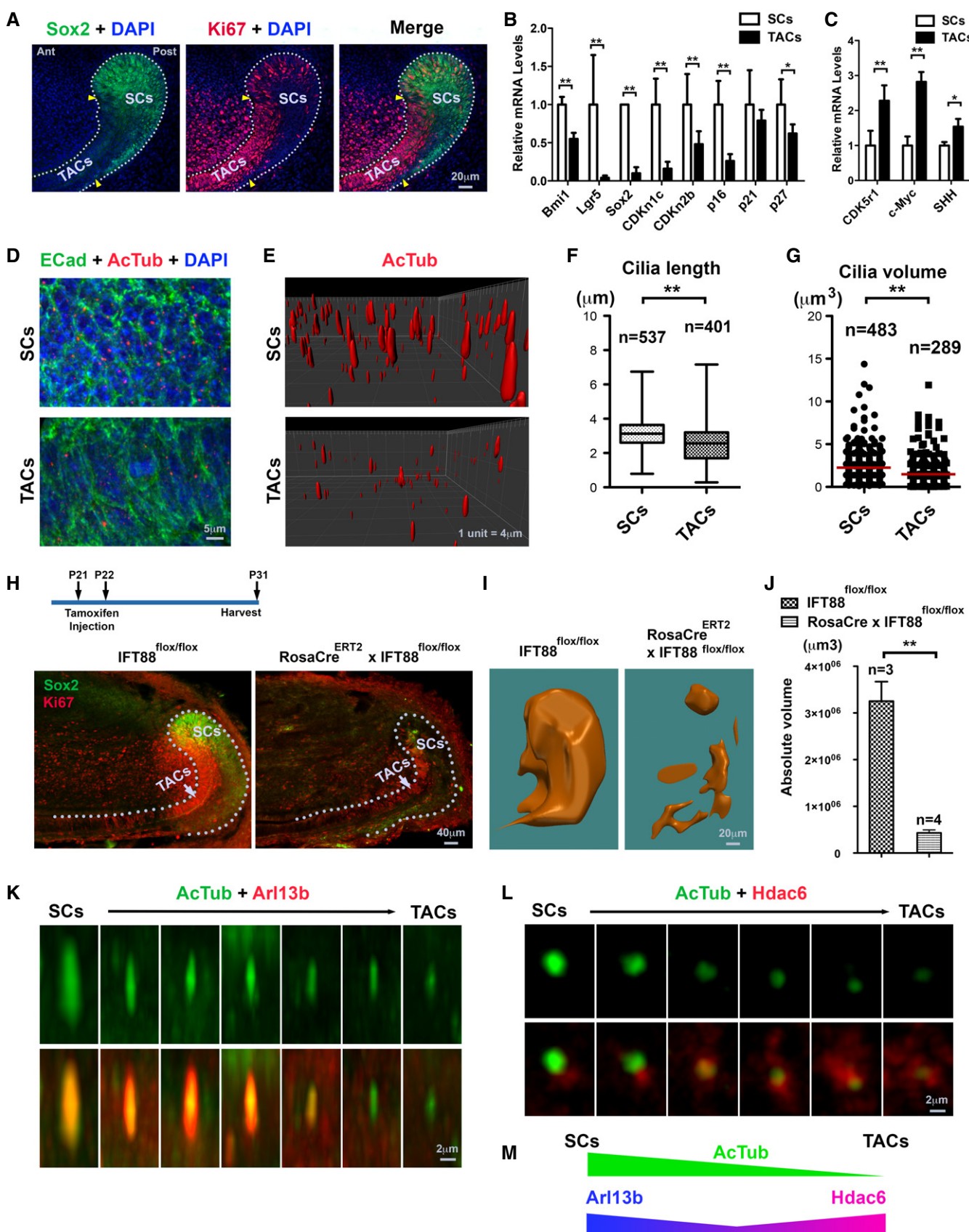

**Figure 1.**

**Figure 1.   Incisor CLE has distinct ciliary dynamics in the stem cells and transit amplifying cells.**

A    Representative IF staining of Sox2 (green) and Ki67 (red) on the P7 CLE stem cell and transit amplifying cell regions and counterstained with DAPI (blue) on a sagittal section. Dotted lines, basement membrane; yellow arrowheads mark approximate stem cell boundaries. SCs, stem cells; TACs, transit amplifying cells; Ant, anterior; Post, posterior.

B, C    The mRNA expression profiling on specific markers of stem cells (B) and transit amplifying cells captured on P7 incisors CLE followed by analysis using real-time RT–PCR (C). qRT–PCR results are in arbitrary values after normalization for *GapDH*. Statistics was performed on triplicates using one-way ANOVA followed by Bonferroni's test. *$P < 0.05$; **$P < 0.01$. Data are presented as mean and standard deviation.

D    Representative IF double staining of E-cadherin (ECad, green, for cell border) and AcTub (red, for the cores of primary cilia) on a sagittal section. Sample is counterstained with DAPI (blue).

E    3D reconstructions of ciliary cores of the stem cells and transit amplifying cells based on AcTub immunostaining.

F, G    Quantification of cilia length (F) and volume (G) based on 3D reconstruction (E) and comparison between those in stem cell and transit amplifying cell regions. Numbers of quantified cilia were indicated (*n*). Statistics was performed using one-way ANOVA. Data are presented as mean and standard deviation. **$P < 0.01$.

H    Tamoxifen-induced IFT88 deletion strategy (top panel) and representative images of Sox2 (green) and Ki67 (red) immunostaining on CLE regions of the indicated mouse strains (bottom panels). Dotted lines, basement membrane. Note the significant reduction of both Sox2 and Ki67 immunolabeling upon deletion of IFT88. For Cre staining on the same set of samples, please see Appendix Fig S1F.

I, J    3D reconstruction of Sox2-positive regions based on its immunolabeling of the indicated mouse strains (I) and their volume comparison between mouse strains as indicated (J). Statistics was performed using Student's *t*-test: **$P < 0.01$. Data are presented as mean and standard deviation.

K, L    Representative IF double staining of ADP-ribosylation factor-like protein 13B (Arl13b, red, K) or histone deacetylase 6 (Hdac6, red, L) with AcTub (green) as a marker of the primary cilium core following stem cell to transit amplifying cell transition.

M    Summary of the immunolabeling of Arl13b or Hdac6 with AcTub (K, L) along stem cell to transit amplifying cell transition.

different elements into the primary cilium. Among them, the ADP-ribosylation factor-like 13B ATPase (hereafter Arl13b) is an unique cilium-associated molecule specifically required for mediating SHH signaling (Mariani *et al*, 2016). In the CLE, we found that Arl13b was preferentially expressed in primary cilia associated with stem cells while its expression disappeared in the transit amplifying cells (Fig 1K and Appendix Fig S1G). On the contrary, when stem cells transitioned into transit amplifying cells, the primary cilia begin to be disassembled and absorbed through deacetylase-mediated tubulin and cortactin deacetylation (Pugacheva *et al*, 2007; Perdiz *et al*, 2011). We observed that histone deacetylase 6 (Hdac6) recruited into primary cilia was particularly evident in the transit amplifying cells (Fig 1L and Appendix Fig S1G). The successive recruitment of Arl13b and Hdac6 therefore represented a dynamic primary cilium biogenesis along the stem cell–transit amplifying cell axis (Fig 1M).

**Prom1 differentially co-expresses with Arl13b at primary cilium during stem cell activation**

One major defect of the *IFT88* mutation is the failure of photoreceptor outer segment assembly and maintenance (Pazour *et al*, 2002). Interestingly, alterations of *Prom1* gene cause similar photoreceptor defects (Zacchigna *et al*, 2009), raising the possibility that Prom1 participates in stem cell fate determination through affecting cilium dynamics. Using two distinct antibodies targeting either the extracellular domain or cytoplasmic C-terminal end of Prom1, we detected an increased Prom1 expression along the stem cell–transit amplifying cell axis in CLE (Fig 2A and Appendix Fig S2A, respectively). From stem cell toward transit amplifying cell transition, Prom1 had an increased cell surface expression on the transit amplifying cells by comparison with stem cells (Fig 2B). The association of Prom1 with primary cilia that are highlighted with AcTub and Arl13b is evident (Fig 2C and Appendix Fig S2B). In addition to primary cilia, Prom1 labels microvilli as we previously demonstrated in other epithelial cells (Weigmann *et al*, 1997; Fig 2B and Appendix Fig S2C). The specificity of Prom1 antibodies was validated using the *Prom1* KO

mice where the respective immunoreactivity was almost abolished (Fig 2D, see below). Likewise, we could again validate the Prom1 antibodies using established primary CLESCs (Appendix Fig S2E, see Materials and Methods) where Prom1 expression (transcript and protein) was silenced by short hairpin RNA (shRNA; Fig 2E and F).

In the cultured CLESCs, the expression of Prom1 is detected in almost of cells with low and high Sox2-positive signals (Fig 2G). In addition to cell surface expression of Prom1 in stem cells and transit amplifying cells, we noticed a nuclear Prom1 immunolabeling in native tissues (Fig 2D and Appendix Fig S2C and D) and in cultured CLESCs (Fig 2G). This prompted us to validate these observations by subcellular fractionation of CLESCs into cytoplasm/membrane (C+M) and nuclear (N) fractions and their immunoblotting. We found a significant fraction of Prom1 immunoreactivity; notably, its full-length species was associated with nuclei and was co-expressed with some cells express low-level Sox2 cells (Fig 2G and H). Similarly, a fraction of Arl13b was detected in the nuclei (Fig 2I). In the mouse embryonic fibroblasts, Arl13b has also been reported being able to translocate into nuclei (Larkins *et al*, 2011). Hence, the dual localization Prom1 and Arl13b suggested that they had cell membrane and nuclear functions involving distinct or coordinated mechanisms, possibly similar to EGF receptor (Lin *et al*, 2001).

**Epithelial Prom1 plays dominant role in the CLE development**

To investigate the impact of Prom1 on stem cells particularly on primary cilia, we analyzed *Prom1* knockout (KO) mice (Zacchigna *et al*, 2009) using IF staining and 3D reconstruction. The total volume of the stem cell region as revealed with Sox2 staining was significantly reduced in the absence of Prom1 (Fig 3A and B). However, cell number for a given surface was unaffected (Appendix Fig S3A). In parallel, the total number of Ki67[+] cells inside the stem cell region was also highly decreased (Fig 3C and D). Hence in the CLE, the *Prom1* KO phenotypes could phenocopy the *IFT88* mutant (Fig 1H–J), suggesting a failure of stem cell activation in the absence of Prom1.

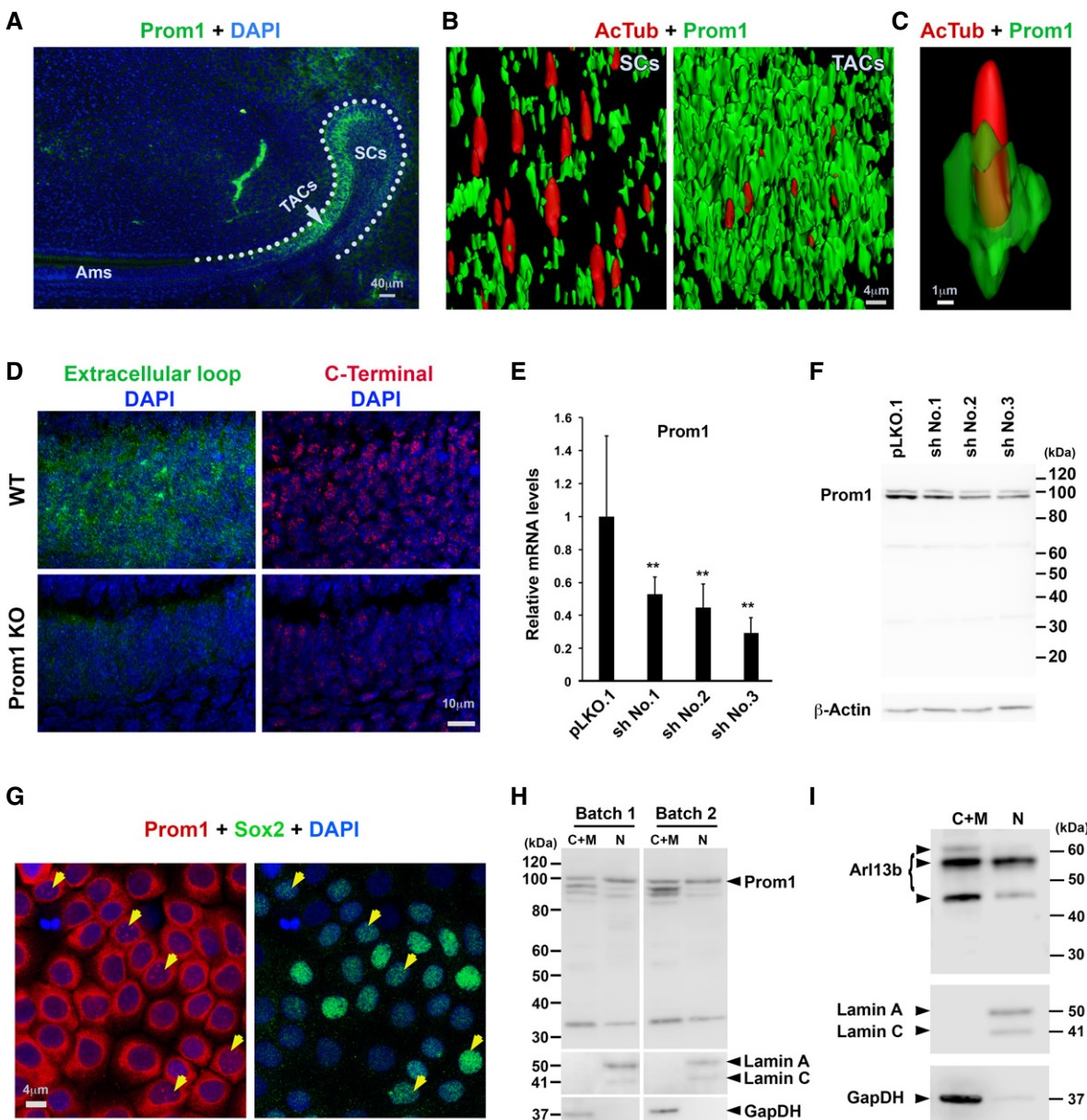

**Figure 2. Prom1 has a dynamic expression in the incisor CLE primary cilia and nuclei.**

A    Representative IF staining of Prom1 using specific antibody clone 13A4 targeting extracellular loop (green) on the stem cell and transit amplifying cell regions of lower incisor CLE at P7. Sample is counterstained with DAPI (blue). Dotted lines, basement membrane. SCs, stem cells; TACs, transit amplifying cells; Ams, ameloblasts.

B    3D reconstruction showing the association of Prom1 (green) with AcTub-labeled (red) primary cilia in stem cell and transit amplifying cell regions. Note that the expression of Prom1 is not limited to primary cilium but also to microvilli.

C    A representative example of Prom1 association with one primary cilium at the stem cell to transit amplifying cell transition region. Green channel transparency was set up to 70%.

D    Representative IF staining of Prom1 using antibodies directed either its extracellular loop (clone 13A4, green) or cytoplasmic C-terminal end (Biorbyt, Orb129549, red) on transit amplifying cell regions of the WT vs. *Prom1* KO mice. Samples are counterstained with DAPI (blue). Note the lack of Prom1 labeling in *Prom1* KO mice.

E, F    The mRNA (E) and protein (F) profiling on shRNA-mediated Prom1 knockdown (3 different shRNAs were used, marked as NO. 1, 2, and 3) in cultured CLESCs. qRT–PCR results are in arbitrary values after normalization for *GapDH*. The immunoblotting was performed with anti-Prom1 C-terminal antibody. β-Actin was used as loading control (F). Molecular mass markers are indicated. Statistics was performed using one-way ANOVA followed by Bonferroni's test: **$P < 0.01$. Data are presented as mean and standard deviation.

G    Representative IF staining of Prom1 (red) and Sox2 (green) in CLESCs *in vitro*. Sample is counterstained with DAPI (blue). Arrows indicate colocalization of Prom1 with low–medium level of Sox2 in the cell nuclei.

H, I    Subcellular fractionation of CLESCs into cytoplasm/membrane (C+M) and nuclear (N) fractions. Materials were analyzed by immunoblotting for Prom1 (H) and Arl13b (I). As controls, membranes were blotted for anti-Lamin A/C (nuclei markers) and GapDH (cytoplasm marker). Molecular mass markers are indicated. Note the presence of full-length Prom1 as well as Arl13b in nuclear fraction.

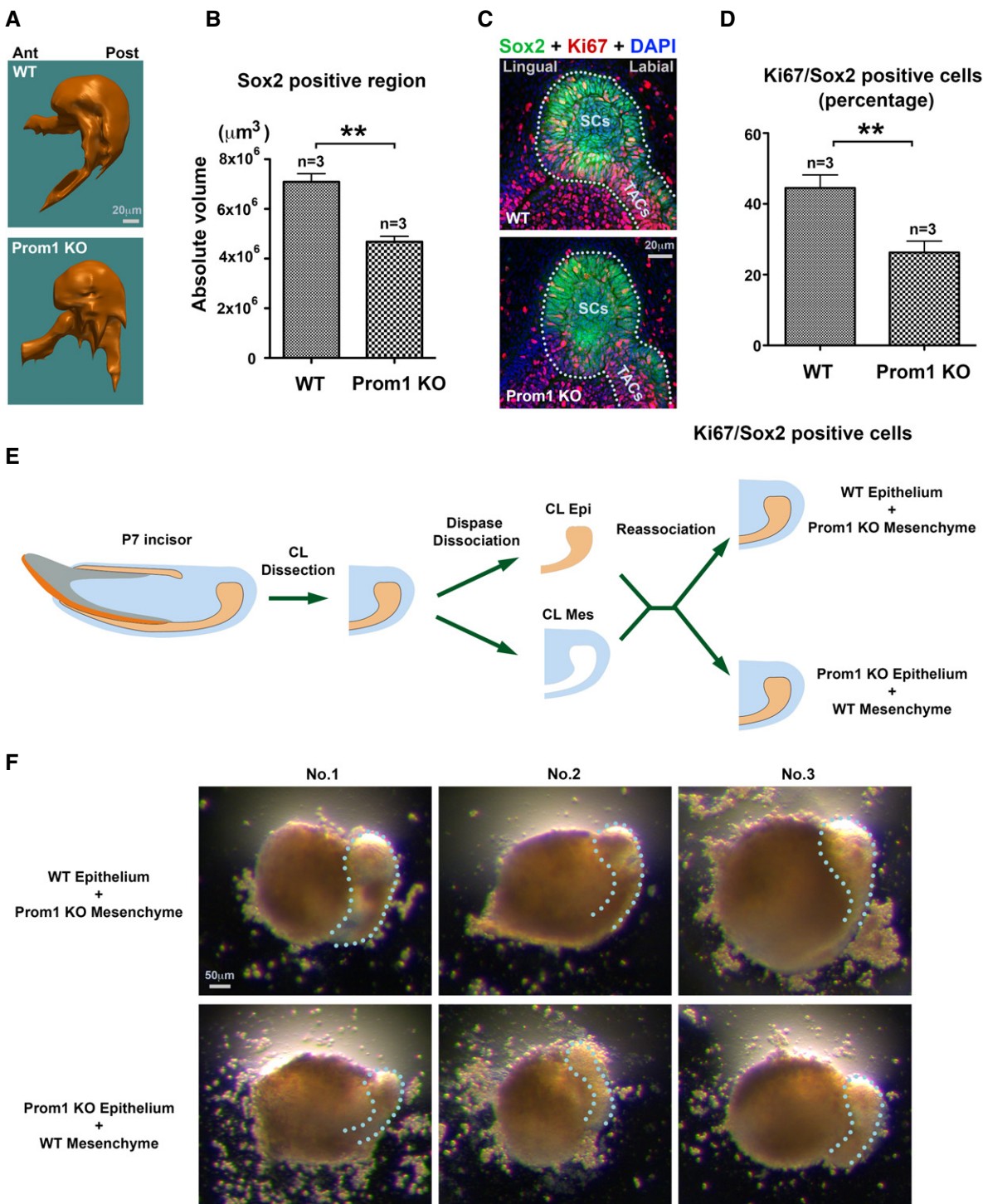

**Figure 3. Epithelial Prom1 regulates CLESC maintenance and activation.**

A, B    Representative images (A) and quantitative analysis (B) of IF staining and 3D reconstruction on Sox2-positive region for WT vs. *Prom1* KO mice. Ant: anterior; Post: posterior (A). Statistics was performed using Student's *t*-test: **$P < 0.01$. Data are presented as mean and standard deviation.

C    Representative IF double staining of Sox 2 (green) and Ki67 (red) on the stem cells and transit amplifying cell regions of WT vs. *Prom1* KO mice on frontal sections. Samples are counterstained with DAPI (blue).

D    Quantification of Ki67[+] cell number in Sox2-positive region based on IF data from WT and *Prom1* KO mice (C). Statistics was performed using Student's *t*-test: **$P < 0.01$.

E    Illustration of CLE tissue recombination assay. P7 CLE was microdissected and then dissociated with Dispase II to achieve epithelium and mesenchyme. The *Prom1* KO epithelium was then reassociated with WT mesenchyme or vice versa.

F    Stereo images of three explants cultured with indicated recombination conditions before harvesting at day 3. Dotted line indicates the CLE.

Prom1 is also expressed in the adjacent mesenchymal cells in front of CLE (Fig 2A and Appendix Fig S2A). A challenging question is in such a tissue whose development relies on epithelial–mesenchymal interactions, whether the Prom1 deletion in the mesenchymal cells could contribute to CLE phenotype, as we have also observed decreased Ki67-positive cells in the *Prom1* KO mice's incisor mesenchyme (Fig 3C). We therefore performed tissue recombination and *in vitro* explant culture assays by switching the epithelium and mesenchyme from the wild-type (WT) and *Prom1* KO mouse incisors (Fig 3E). The results showed that the *Prom1* null epithelium more likely contributed to the smaller CL (Fig 3F) and less Sox2 and Ki67 expression, rather than the mesenchyme (Appendix Fig S3B). Collectively, we confirmed Prom1 in the CLE had a dominant role in controlling CLE tissue growth.

### Prom1 is required for stem cell maintenance and activation

We next evaluated whether the reduced volume of the stem cell region was due to a loss of self-renewal capabilities in *Prom1* KO stem cells. Using CLESCs, we confirmed that while WT cells could be passaged for at least 15 generations, the primary *Prom1* KO mouse-derived cells could not be maintained for more than a single passage (Fig 4A). As noted above, SHH is one key signaling pathway required by stem cell survival and self-renewal in many systems. We found SHH-stimulated cell proliferation (e.g., colony formation) was almost abolished in the primary CLESCs derived from *Prom1* KO mice (Fig 4B and C, and Appendix Fig S4A). Similarly, siRNA-mediated *Prom1* knockdown also highly decreased CLESC clonogenic capability (Appendix Fig S4B–E). To test the CLESCs' spheroid formation capabilities, we mixed CLESCs with the CLE mesenchymal cells at ratio of 1:1 with/without knockdown of Prom1 in each cell type. The results showed that the mixed epithelial–mesenchymal cells generated smaller spheres only if epithelial *Prom1* was knocked down (Fig 4D and E); hence, again we confirmed that in the CLE region, the epithelial Prom1 had the predominant role.

To investigate how *Prom1* controlled stem cell activation particularly through regulating quiescent cells' cycle re-entrance, we engineered a modified fluorescence ubiquitination-based cell cycle indicator (FUCCI) system (Sakaue-Sawano *et al*, 2008; Zambon, 2010) in which the expression of FUCCI was driven by a *Ki67* promoter (Fig 5A). With the system, we confirmed stem cells (in G0 phase, i.e., quiescence) could be identified as colorless cells (Fig 5B) and Prom1 overexpression in the CLESCs significantly increased the proportion of cells at G0 and G2/M phases (Fig 5C and Appendix Fig S5A), similar to the effect of SHH treatment (Appendix Fig S5B), suggesting that Prom1 was indeed critical for stem cell self-renewal and proliferation. In parallel, only in the presence of SHH, we found the cell ciliary dynamics were correctly associated with the cell cycle (Fig 5D–F; Sanchez & Dynlacht, 2016). Collectively, these lines of evidence suggested that Prom1 was indeed an important mediator of SHH signaling in stem cell activation and renewal.

### Arl13b and Hdac6 compete binding site to Prom1

The failure of stem cell maintenance and renewal upon loss of *Prom1* was possibly due to the inability of stem cells in responding to SHH. To gain insight into how Prom1 mediated the SHH signal, we analyzed the properties of primary cilium structures

and found that their length was reduced in *Prom1* KO stem cells but increased in transit amplifying cells revealing that their dynamics between stem cells and transit amplifying cells were perturbed (Fig 6A). In agreement with that, the primary cilium density was significantly increased in both regions in the *Prom1* KO mice (Fig 6B). Since we have observed successive recruitment of Arl13b and Hdac6 to primary cilia during stem cell to transit amplifying cell transition (Fig 1K and L), we therefore analyzed whether Hdac6 and Arl13b could possibly be regulated by Prom1 through a similar mechanism and could be affected in the absence of Prom1. In line with such hypothesis, we found that Hdac6 inhibitor, tubacin, and silencing Arl13b siRNA on CLESCs could reduce colony formation (Fig 6C and D, respectively). Similarly, siArl13b or siHdac6 treatments mirrored the effects of blocking stem cell activation by increasing the cell number stacked at G0 phase (Fig 6E–G).

To investigate whether Prom1 directly modulates primary cilium size, we used a Prom1 mutant in which lysine 138 (in the Hdac6-binding site, see below) was mutated to glutamine (K138Q; Fig 6H). This mutation neutralizes the electric charge and mimics lysine acetylation. We found that the ectopic expression of Prom1 in polarized epithelial MDCK cells led to a significant increase in the primary cilium size while using the K138Q mutant, the overexpressing Prom1's effects on primary cilia size were impaired, and the number of Prom1$^+$ cells without primary cilium was increased as observed upon IF staining for Prom1 and AcTub and confocal laser scanning microscopy and scanning electron microscopy (SEM) analyses (Fig 6I and J and Appendix Fig S6A–C). As expected, a reduction of AcTub immunoreactivity and an increase of the Hdac6/AcTub ratio were observed in cilia of cells expressing K138Q mutant (Appendix Fig S6A, D and E). These effects could not be reversed by adding tubacin (Appendix Fig S6F and G) in line with an earlier report (Haggarty *et al*, 2003).

It has been reported the binding of Hdac6 to Prom1 through its lysine 138 (Mak *et al*, 2012), which raised the possibility that Prom1 as a regulator of ciliary architecture binds as well to Arl13b. Remarkably, we observed that Prom1 is co-immunoisolated with Arl13b (as well the reciprocal experiment) and this interaction is impaired upon mutation K138Q (Fig 6K–M). Consistently in the CLESCs, knockdown of Prom1 reduced Arl13b expression both in the cytoplasm and in the nuclear fractions (Appendix Fig S6H and I). We also observed *in vivo* that Arl13b staining was diminished, if not absent, from most of the primary cilium in *Prom1* KO mice (Appendix Fig S6J).

Therefore, Prom1 mediated Hdac6 and Arl13b recruitment to the primary cilium, possibly through a competitive mechanism. As such, in the absence of Prom1, SHH signal transduction in the primary cilium was inhibited: Although the axoneme was still present, the ability of the primary cilium to respond to SHH was lost.

### Lineage differentiation is compromised in the *Prom1* KO epithelial cells

In MDCK cells, we observed that Prom1 K138Q mutation enhances the dome formation, but significantly reduces cell proliferation (Fig 7A–D). Moreover, the remaining longer cilia in MDCK cells incorporated with K138Q mutation appeared with a "pearling" morphology as observed by SEM suggesting an enhanced release of membrane vesicles (Fig 7E). We previously demonstrated that the

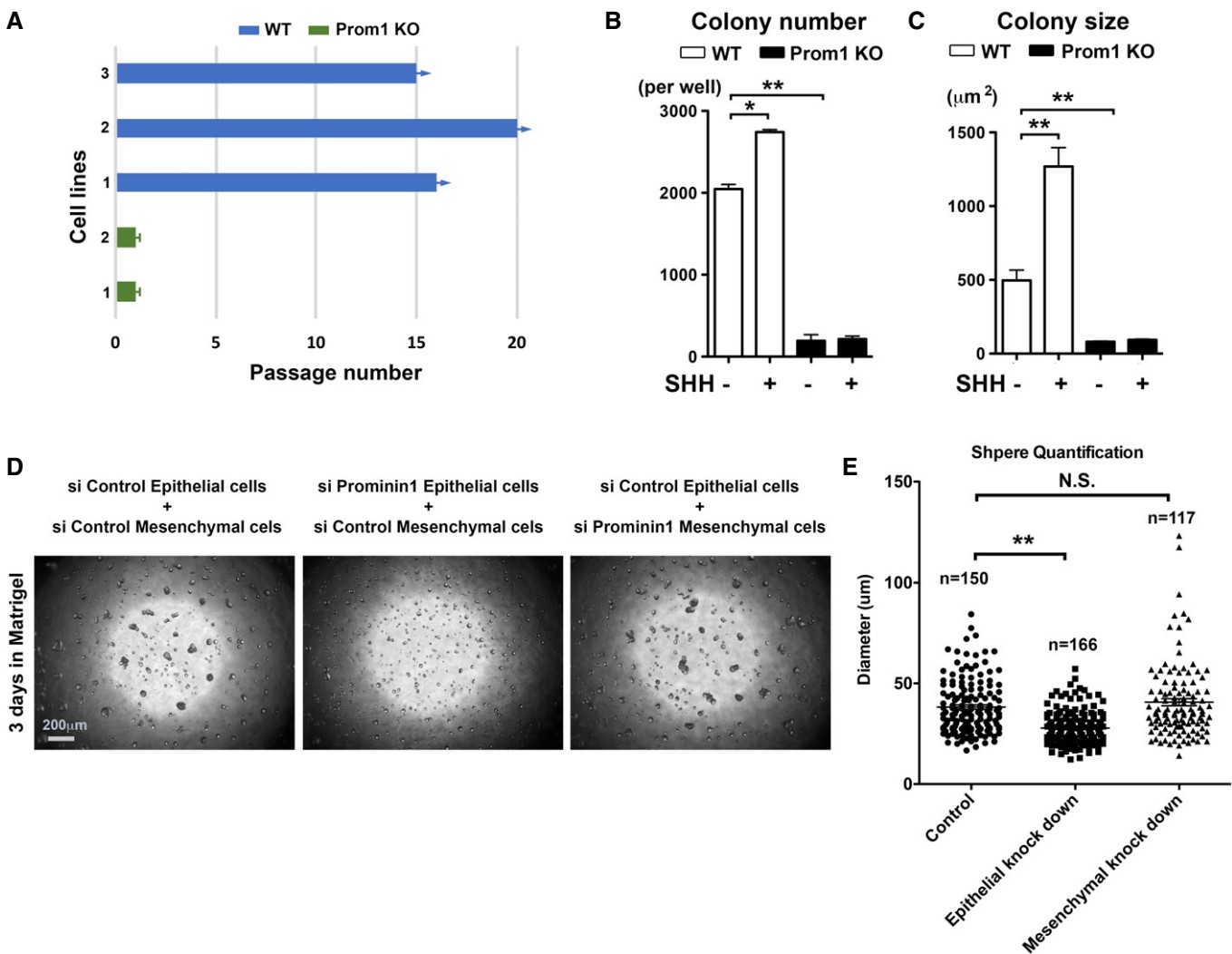

**Figure 4. CLESC renewal is compromised in the absence of Prom1.**

A Passaging capacity analysis of the CLESCs derived from P7 WT and *Prom1* KO mice. Note the passages of cells derived from WT animals are still ongoing while those from *Prom1* KO mice were arrested at the first passage. Arrows indicate cells that were still under passaging, while blunt bars represent cells that could not be passaged and stopped at the indicated passage.

B, C Clonogenicity assay for colony number (B) and size (C) after the first the passage of CLESCs derived from WT vs. *Prom1* KO mice in the absence or presence of SHH (100 ng/ml). Cells were seeded in triplicate on 24-well plates as 5,000 cell/per well. Statistics was performed using Student's *t*-test: *$P < 0.05$; **$P < 0.01$. Data are presented as mean and standard deviation of triplicated samples.

D Sphenoid analyses. For these assays, epithelial and mesenchymal cells were mixed at ratio of 5,000:5,000 per well in triplicates on 96-well plates and grown for 3 days on Matrigel. Representative micrographs of entire well under indicated conditions are shown.

E Quantification of sphere diameter on samples presented in (D). Statistics was performed using Dunnett's test: **$P < 0.01$. N.S., non-significant. Number of quantified spheres is indicated (*n*).

release of Prom1[+] vesicles is associated with stem cell differentiation (see Discussion). The differential centrifugation of conditioned media followed by the immunoblotting of recovered cellular and vesicle fractions revealed the amount of Prom1 protein found in the vesicle fraction was significantly increased in K138Q mutant by comparison with non-mutated Prom1 (Fig 7F and G). Thus, the reduction of AcTub observed in primary cilia of K138Q mutant-expressing cells might result in the fission of Prom1[+] vesicles concomitant with the reduction of ciliary length. Besides a direct impact of cell fate, these observations might also negatively affect *in vivo* cell secretion and crosstalk between different cell types, such

as stem cells and transit amplifying cells, as well as epithelial–mesenchymal interactions.

Ameloblasts are CLE's stem cell-derived functional differentiated cells that have high secretion activity. To decipher the biological consequences of losing Prom1 on ameloblasts, we analyzed teeth from 2-month-old *Prom1* KO mice. Macroscopic and microscopic analyses showed developmental defects affecting tooth enamel in *Prom1* KO mice (Appendix Fig S7A–C) that can be explained by the abnormal aggregation of vesicles in ameloblasts (Appendix Fig S7D) with reduced lineage tooth epithelial differentiation marker expression (Appendix Fig S7E). Intriguingly,

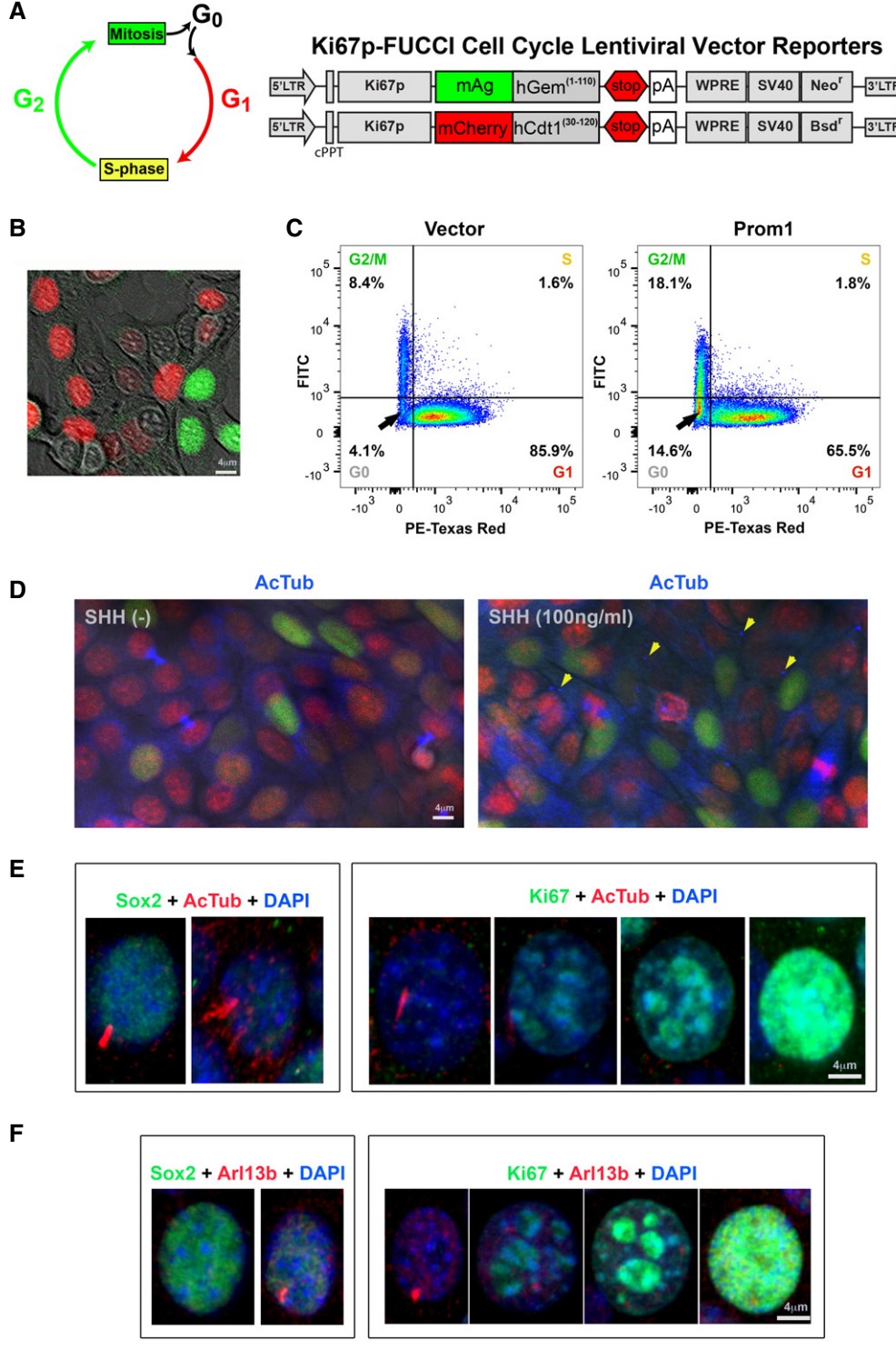

**Figure 5.  Prom1 is important for quiescent cell re-enter cell cycle.**

A   Designing strategy of the Ki67-FUCCI cell cycle indicator system.

B   Representative image of CLESCs harboring the Ki67-FUCCI system showing colored cells at different cell cycle phases.

C   Flow cytometry analysis of Ki67-FUCCI CLESCs at different cell cycle stages in cells transfected with empty vector or full-length Prom1. As a result, > 95% of the events have been included in the analysis. The accuracy of the strategy has been validated by two independent researchers for IF quantification using Fiji 1.0 software.

D   Representative IF staining of AcTub (blue) on Ki67-FUCCI CLESCs without and with the presence of SHH. Note that the cilium could be visualized (arrows) solely when SHH is present.

E, F   Representative IF double staining of either AcTub (D, red) or Arl13b (E, red) with Sox2 (green) or Ki67 (green) in CLESCs in the presence of SHH. Note the expression levels of Sox2 and Ki67 distinguished stem cells and transit amplifying cells, respectively.

            

**Figure 6.  Prom1 regulates ciliary dynamics by directly interacting with Hdac6 and Arl13b.**

A, B    Size of individual cilia in stem cell and transit amplifying cell regions of WT and *Prom1* KO mice (A) and their density in the respective regions (B). Three animals were analyzed for each genotype. The number of quantified cilia is indicated (*n*). Statistics was performed using Dunnett's test: \**P* < 0.05; \*\**P* < 0.01. Data are presented as mean and standard deviation. SCs, stem cells; TACs, transit amplifying cells.

C, D    Clonogenicity analysis of CLESCs treated with either Hdac6 inhibitor (tubacin) and appropriate controls (niltubacin and DMSO, C) or siArl13b and its siControl. Colonies were stained with crystal violet (D).

E, F    siRNA knockdown efficiency of Arl13b (E) and Hdac6 (F) in Ki67-FUCCI CLESCs was evaluated by real-time RT–PCR analysis. The results are in arbitrary values after normalization for *GapDH*. Statistics was performed in triplicates using one-way ANOVA followed by Bonferroni's test: \*\**P* < 0.01. Data are presented as mean and standard deviation.

G    Flow cytometry analysis and quantification of samples indicated in (E and F).

H    Mutation strategy of Hdac6-binding site on Prom1. The lysine (K) 138 in the first cytoplasmic loop of Prom1 was mutated to glutamine residue (Q). TM, transmembrane domain.

I    Representative IF double staining and 3D reconstruction of the Prom1 (green) and AcTub (red) at the apical domain of polarized MDCK cells expressing Prom1 and its K138Q mutant (top panels). Black-white inserts show Prom1 signals only. Same cells were analyzed by SEM (bottom panels). Note that Prom1 is asymmetrically distributed along the primary cilium (white arrows) as observed *in vivo* (see Fig 2C) and mutation K138Q significantly impacts on the primary cilium length. MV, microvilli.

J    IF staining and 3D reconstruction of AcTub-labeled (red) primary cilium at the apical membrane of parental MDCK cells (MDCK) and those cells transfected with Prom1 or its K138Q mutant. The images were used to quantify the primary cilium length (Appendix Fig S6B and C). Arrows indicate primary cilia with more than 4 µm in length.

K–M    Co-immunoisolation (IS) analyses were performed with paramagnetic beads using antibodies either against Arl13b (K) or Prom1 (AC133) or with anti-FITC antibody as control (L). Recovered materials were analyzed by immunoblotting against the indicated proteins. Brackets and asterisks indicate the plasma membrane and endoplasmic reticulum-associated forms of Prom1, respectively, while arrows show Arl13b. Arrowhead, immunoglobulin. The ratio Arl13b/Prom1 immunoreactivities was quantified (*n* = 3) (M). Mann-Whitney test: \*\**P* < 0.01. Data are presented as mean and standard deviation.

the phenotypes resemble some, if not all, of the tooth phenotypes of *Wnt* (Millar *et al*, 2003) and *Notch* mutant mice (Hu's group unpublished data). The detailed mechanisms responsible for Prom1 function in ameloblast lineage differentiation will be reported separately.

### Glis2 is a partner of Prom1 at primary cilium and nucleus in mediating SHH signaling activation

The above observations encouraged us to investigate the downstream consequences of SHH signal transduction failure in stem cells. To identify which SHH downstream effectors such as Gli or Glis transcriptional factors were more prone to be a working partner of Prom1, we profiled their expression in the CLE. We found only Glis2's transcript is specifically elevated in the CLE's transit amplifying cells (Fig 8A). Using specific antibodies targeting Gli1 (Zhao *et al*, 2014) and Glis2, we confirmed that while Gli1 protein was expressed in almost all the CLE cells and ameloblasts, Glis2 expression is mainly restricted to transit amplifying cells and to less extent in stem cells (Fig 8B and Appendix Fig S8A and B). Glis2 expression was identified in two distinct subcellular compartments: the nuclei of transit amplifying cells (Fig 8C) and primary cilia during stem cell to transit amplifying cell transition (Fig 8D), confirming its potential translocation route from primary cilia to nuclei.

We next examined the consequence of losing Glis2 *in vivo*. In the *Glis2* KO mice, we observed a significant reduction of the volume of stem cell Sox2-positive region (Fig 8E), similar to the *Prom1* KO mice (Fig 3A and B). Strikingly, the Prom1 expression was almost abolished in the CLE of the *Glis2* KO mice (Fig 8F), implicating the high possibility of Glis2 working as a partner of Prom1. Distinguishably, we found primary cilium localization of Glis2 was lost in most cells in *Prom1* KO animals both at their stem cell and at their transit amplifying cell regions (Fig 8G). Glis2 is one of the transcriptional factors of SHH pathway. In the *Prom1* KO CLE, we found that SHH itself, as well as its

downstream targets, Gli1, 2, and 3, was all down-regulated (Appendix Fig S8C and D) and transfecting K138Q Prom1 mutant into CLESCs showed significantly reduced expression of Gli1-3 and Glis1 and Glis2 comparing to full-length Prom1 (Appendix Fig S8E). To confirm whether losing Prom1 or Glis2 has similar effect on incisor homeostasis, we performed microCT analysis on adult mice and could substantiate that both mutant strains developed thinner and shorter enamel layers by comparison with WT animals (Appendix Fig S8F). Together, our data suggested that Glis2 could possibly function as a partner of Prom1 on the ciliary membrane, and even in the nuclei in affecting stem cell homeostasis and activation.

### Stat3 is a direct downstream target of Prom1-mediated Glis2 signaling

To investigate the potential linkage between Prom1 and Glis2, we performed a proximity ligation assay. We found that the two proteins were physically closely associated, and the complex they formed was identified in the nuclei in both stem cells and transit amplifying cells but more prominently in the stem cells (Fig 9A). In the cultured CLESCs, we could further demonstrate that the Prom1-Glis2 complex entered into nuclei in a time-dependent manner and the translocation could be blocked by importazole, a small molecule inhibitor of the function of importin β1 which alters its interaction with Ran-GTP, and hence the nuclear import (Soderholm *et al*, 2011; Fig 9B).

To investigate the signaling pathways that might be regulated by Prom1 and/or Glis2, we analyzed CLE cells in *Prom1*- and *Glis2*-KO mice using a predesigned pathway screening PCR array analysis, and found the two major pathways commonly affected in the two different mutants were p53 and Stat (Fig 9C–E and Appendix Figs S9A and B). Stat3 is a key transcription factor involved in stem cell maintenance and activation, although little was previously known about its upstream regulation and function (Tierney *et al*, 2014). Stat3 protein was highly expressed in the stem cells by comparison

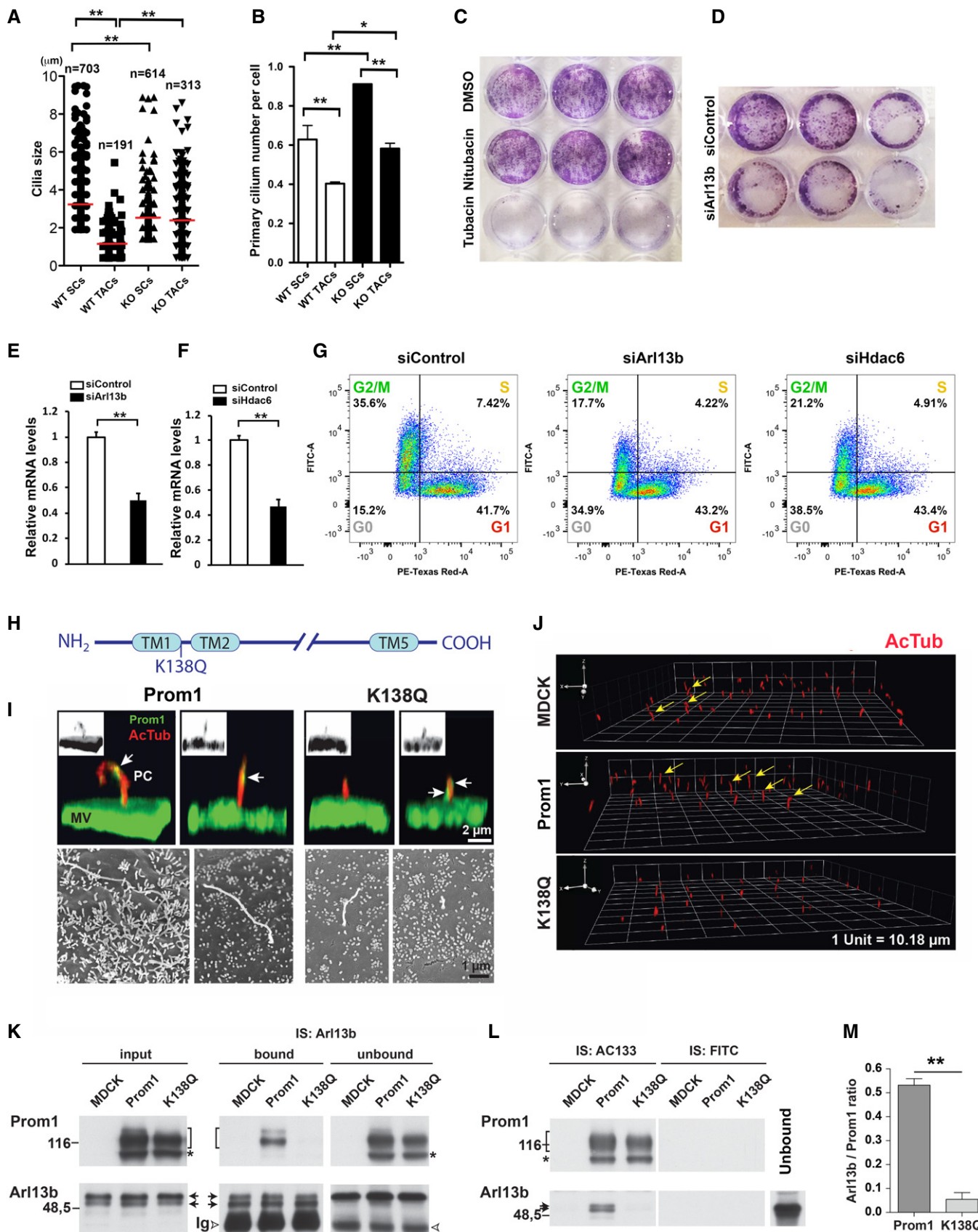

**Figure 6.**

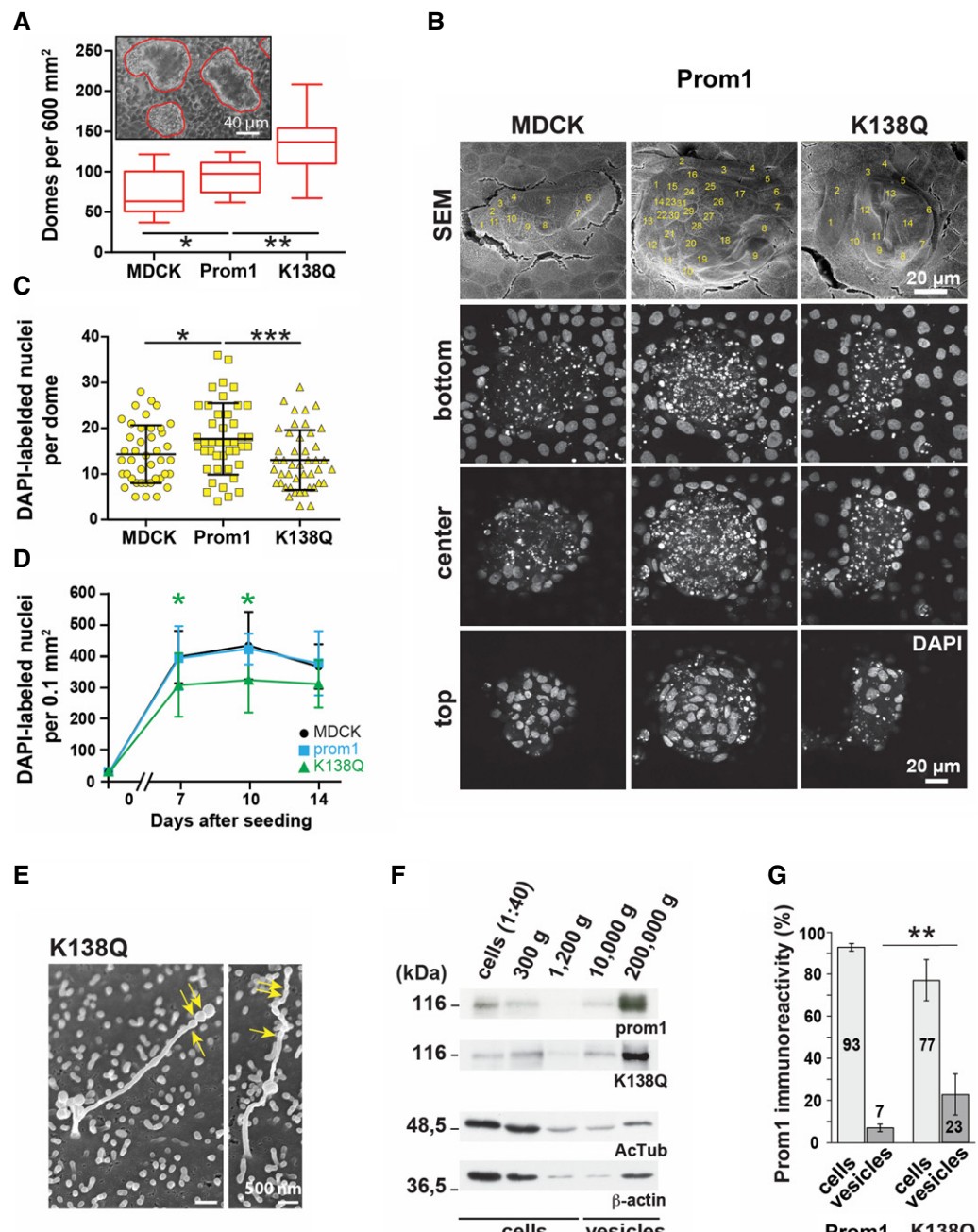

**Figure 7. Prom1 regulates stem cell homeostasis and differentiation through lysine 138.**

A–D  Parental MDCK cells or those transfected with Prom1 or K138Q mutant were grown for 7 dpc (A–C) or as indicated (D). Cells were observed by phase-contrast microscopy (A), SEM (B, top panels), or confocal laser scanning microscopy after DAPI staining (B, bottom panels, C, D). Box–whisker plots show the number of domes per 600 mm² (A). Three images were analyzed per experiment (n = 4). Individual cells within a given dome are indicated (B, top panels) and were quantified using three different planes (bottom, center, and top) of DAPI-labeled nuclei (B, bottom panels). Domes of similar size were evaluated (C). Means and standard deviations are displayed. Fifteen domes per independent experiment were examined (n = 3). Cell density at a given time of culture (7, 10 and 14 days) was determined by counting DAPI-labeled nuclei per 0.1 mm² (D, n = 4). Two-tailed unpaired Student's t-test: *P < 0.05; **P < 0.01. ***P < 0.005.

E  Cilia with a length of > 4 μm were rarely observed in K138Q mutant and displayed an irregular shape with constrictions along their distal part as observed by SEM (arrows).

F, G  Mutation in the Hdac6-binding site of Prom1 increases the release of Prom1[+] vesicles. 16-h conditioned medium was subjected to differential centrifugation for 5 min at 300 × g (P1); 20 min at 1,200 × g (P2); 30 min at 10,000 × g (P3); and 1 h at 200,000 × g (P4). In parallel, detergent cell lysates were prepared. The entire pellets and 1/40 of cell lysates were analyzed by immunoblotting using antibodies against human Prom1, AcTub, and β-actin (F). Molecular mass markers are indicated. All three proteins were detected in the vesicle fractions (P3 and P4), suggesting Prom1[+] vesicles are released from primary cilia and microvilli. The release of Prom1[+] vesicles was quantified using its immunoreactivities associated with cell and vesicle fractions (G) (n = 3). Two-tailed unpaired Student's t-test: **P < 0.01. Data are presented as mean and standard deviation.

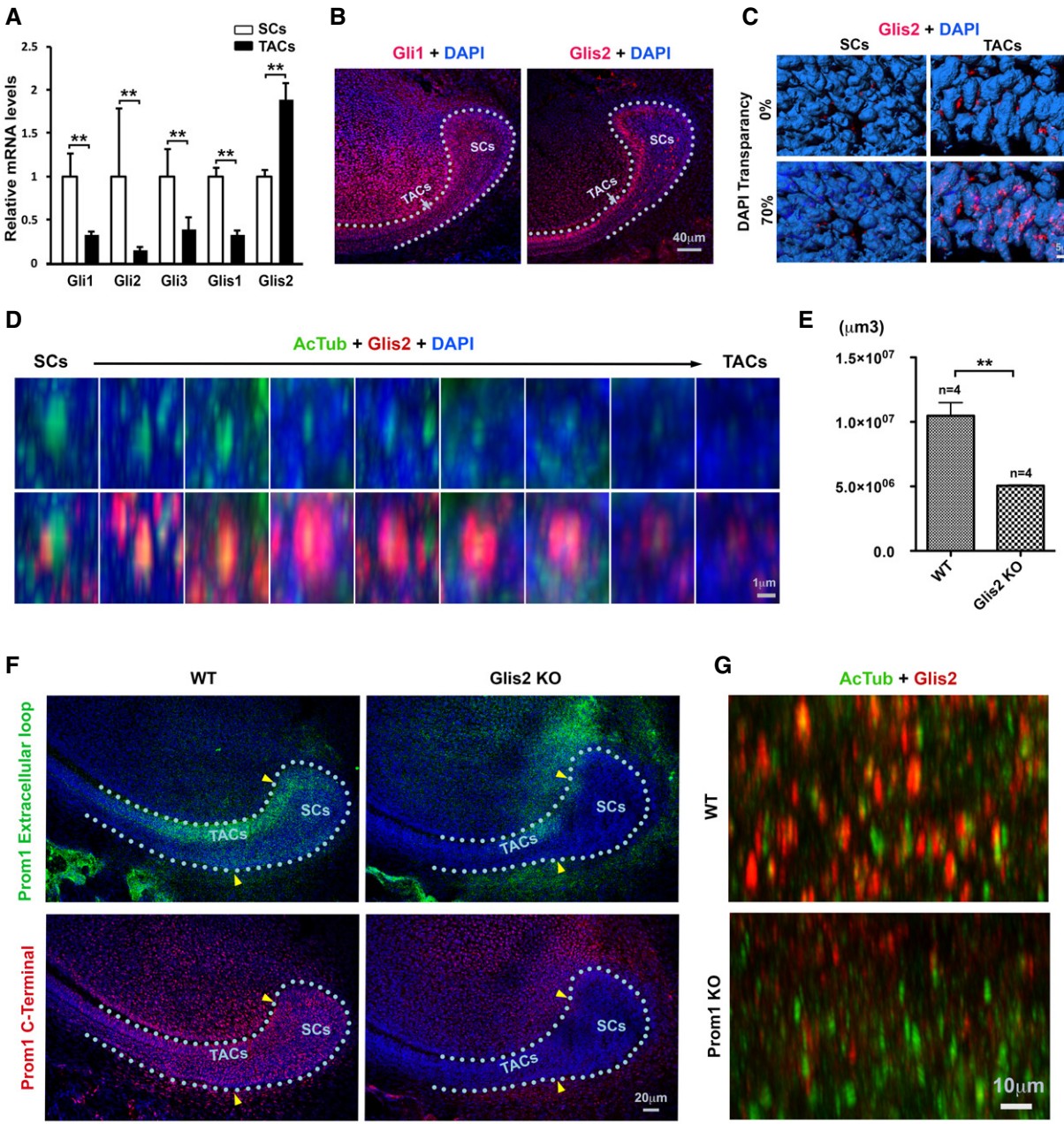

**Figure 8.  Prom1 mediates SHH activation through Glis2.**

A   The mRNA expression of SHH effector genes (Gli1/2 and Glis1–3) on laser microdissection captured samples was evaluated by real-time RT–PCR analysis. The results are in arbitrary values after normalization for *GapDH*. Each sample was isolated from distinct mice (*n* = 3). Statistics was performed using Student's *t*-test: **$P < 0.01$. SCs, stem cells; TACs, transit amplifying cells.

B   Representative IF staining of Gli1 (left panel, red) and Glis2 (right panel, red) on the stem cell and transit amplifying cell regions of lower incisor CLE at P7. Samples were counterstained with DAPI (blue). Dotted lines, basement membrane.

C   3D reconstruction of Glis2 staining (see B, red) and its association with DAPI-labeled nuclei (blue). Nuclear transparency was set at 0 or 70% as indicated.

D   Representative IF double staining of Glis2 (red, bottom panel) and AcTub (green, top and bottom panels) along stem cell–transit amplifying cell axis in lower incisor CLE at P7. Samples were counterstained with DAPI (blue). Note the association of Glis2 with primary cilia particularly in the transition zone between stem cell and transit amplifying cell regions.

E   Volume of Sox2-positive region on the lower incisor CLE at P7 was evaluated in WT vs. *Glis2* KO mice upon IF staining. Number of animals analyzed is indicated (*n*). Statistics was performed using Student's *t*-test: **$P < 0.01$. Data are presented as mean and standard deviation.

F   Representative IF staining of Prom1 using specific antibodies targeting either its extracellular loop (top panels, green) or cytoplasmic C-terminal end (bottom panels, red) on the stem cell and transit amplifying cell regions of lower incisor CLE at P7 WT and *Glis2* KO mice. Samples are counterstained with DAPI (blue). Dotted lines, basement membrane. Arrowheads mark approximate boundary of stem cell regions.

G   Representative IF double staining of Glis2 (red) and AcTub (green) as a marker of primary cilia in stem cell region of WT vs. *Prom1* KO mice. Note the reduction of Glis2 expression in Prom1-deficient animals.

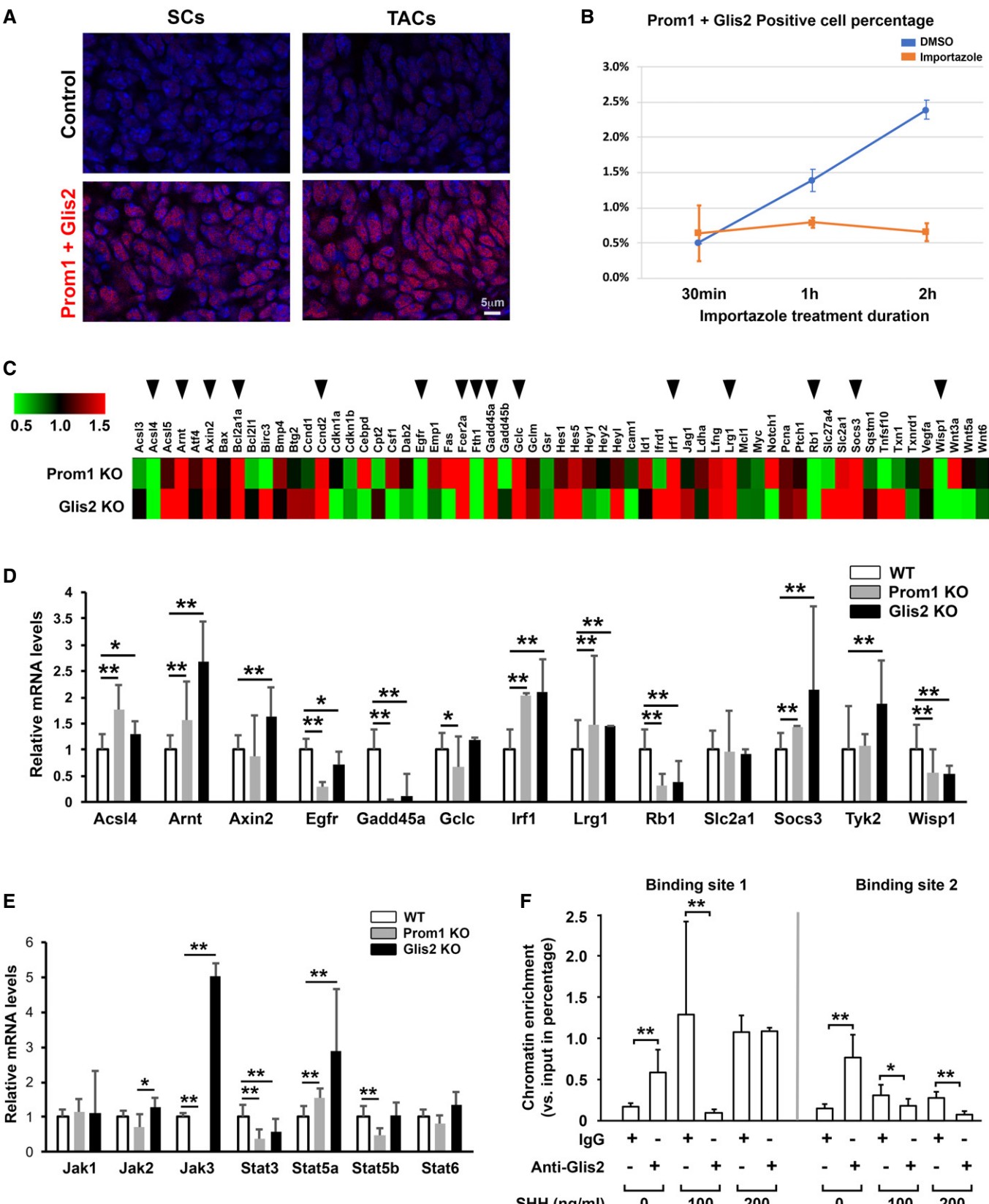

Figure 9.

◀

Figure 9. Stat3 is downstream target of Glis2.

A    Proximity ligation assay analysis of Prom1 and Glis2 association (red) in stem cell and transit amplifying cell regions of lower incisor CLE at P7. As negative control, one conjugated primary antibody (anti-Glis2) was omitted. SCs, stem cells; TACs, transit amplifying cells.
B    Time-dependent Prom1 and Glis2 co-expression (proximity ligation assay) in the cultured CLESCs starved for 4 days, and then treated with SHH (100 ng/ml) and importazole (40 μM), an inhibitor of importin β1 function. DMSO was used as vehicle control. Data are presented as mean and standard deviation of triplicated samples.
C    PCR array analyses were performed on LCM captured P7 CLE of *Prom1* and *Glis2* KO vs. WT mice (*n* = 3). Arrows indicate genes with expression changed for more than 2 folder increase or decrease.
D, E    The mRNA expression of gene significantly changed in the PCR array (in (C)) (D) and Stat pathway members (E) was evaluated in WT, *Prom1*, and *Glis2* KO mice by real-time RT–PCR analysis. Each sample was isolated from distinct mice (*n* = 3). The results are in arbitrary values after normalization for *GapDH*. Statistics was performed using one-way ANOVA followed by Bonferroni's test: $*P < 0.05$; $**P < 0.01$. Data presented as mean and standard deviation.
F    Chromatin IP analysis of binding of Glis2 to Stat3 promoter (see Appendix Fig S9D) in the absence or presence of SHH at different concentration as indicated on triplicated samples, using specific anti-Glis2 antibodies and rabbit IgG as control. Statistics was performed using one-way ANOVA followed by Bonferroni's test: $*P < 0.05$; $**P < 0.01$.

with transit amplifying cell region (Appendix Fig S9C). By performing chromatin IP analysis on the potential binding sequence on the mouse Stat3 promoter in CLESCs, we confirmed that Stat3 was a direct transcriptional target of Glis2 (Fig 9F and Appendix Fig S9D). Therefore, our studies have revealed the upstream components of the Stat3 signaling pathway. It is known that losing Stat3 (Nam *et al*, 2005; Al Zaid Siddiquee & Turkson, 2008) and losing Glis2 (Attanasio *et al*, 2007) both induce apoptosis in different systems. In *Prom1* KO CLE and siProm1-treated CLESCs, we also observed a significant increase in cell apoptosis (Appendix Fig S9E and F) and elevated active caspase-3 expression (Appendix Fig S9G). Collectively, our data suggested that Prom1 is the key coordinator of SHH signaling that controls stem cell maintenance and activation, through interacting with Glis2 and regulating downstreams such as Stat3.

**Prom1 is a responding element of epigenetic cues**

Stem cell renewal and differentiation require a proper response to external signals, which relies on rigidly controlled epigenetic programs (Cheung & Rando, 2013; Rinaldi & Benitah, 2015). In this context, we found that Prom1 is directly regulated by a Polycomb group protein, i.e., Bmi1. In the *Bmi1* KO mice, Prom1 expression was strongly deregulated in the CLE (Appendix Fig S9H), which resulted in a significant reduction in the number of cilia and their length (Appendix Fig S9I and J) as well as Glis2 expression (Appendix Fig S9K). Collectively, our observations supported a central role for Prom1 in mediating stem cell activation.

# Discussion

By dissecting the molecular switch regulating incisor CLE stem cell turnover, we demonstrated that Prom1 functions as a key recruiter of ciliary regulators in a cell status-dependent manner: cilia-responding element Arl13b in quiescent stem cells and Hdac6 at the transient amplifying stage. Given the prevalence of Prom1 expression in different stem cell types and various differentiated cells, we do not exclude that Prom1 might have other molecular partners. One excellent candidate might be downstream of Hedgehog (HH) signaling, as HH is the best-known system requiring primary cilia to coordinate their transduction. The Gli family functions as transcriptional factors of the HH pathway (Goetz & Anderson, 2010). In the other systems, Gli1 and Gli2 could be released from primary cilia upon SHH activation (Goetz & Anderson, 2010). In this study, we

reported Glis2 could perform similar nuclear translocation together with Prom1 and share some common downstream targets, demonstrating Prom1 might have additional unexpected cilia-to-nuclear functions that require further study.

We have observed that *in vitro*, consistent with findings in the cancer cell lines (Damek-Poprawa *et al*, 2011; Waldron *et al*, 2011), losing Prom1 has severe consequences on the survival of CLESCs. *Prom1* null cells abolished cell cycle regulation and entered apoptosis. However, *in vivo*, in the *Prom1* KO mouse CLE, some stem cells could still progress into transit amplifying cells, possibly through bypassing or compensation mechanisms. Other prominin family members have been discovered (Prom2, Prom3; Fargeas *et al*, 2003; Han & Papermaster, 2011), which are also expressed on cellular membrane protrusions (Florek *et al*, 2007), with or without the presence of Prom1. For instance, zebrafish Prom3 is expressed in certain tissues and/or organs in the absence of Prom1 (e.g., József Jászai and D.C., manuscript in preparation), and highly relevant for our study, the murine Prom2 is co-expressed at the CLE transit amplifying cells (Seidel *et al*, 2017). The neural stem cells from adult murine hippocampus are a good example illustrating a compensatory mechanism where an upregulation of Prom2 is observed in the dentate gyrus of Prom1-deficient animals (Walker *et al*, 2013). Therefore, it remains to be determined the role of prominin paralogues on cell fate as well as their impact on plasma membrane protrusions.

One surprising observation in our study is the translocation of full-length Prom1 (as other proteins) into the nuclear compartment. This finding is nonetheless in line with publications describing the presence of Prom1 in the nucleus of cells, which might be an indicator of poor prognosis in cancer (Cantile *et al*, 2013; Huang *et al*, 2015). How can a plasma membrane protein reach the nucleoplasm? First, our data with importazole treatment suggest that nuclear pores are involved in this process instead of the nuclear entrapment following the reorganization of the nuclear envelope during mitosis. Second, novel intracellular routes involving endosomal pathways were recently described to explain the nuclear localization of surface proteins (Chaumet *et al*, 2015; Santos *et al*, 2018). One of them relies on the presence of late endosomes in the nucleoplasmic reticulum where the endocytosed membrane proteins including Prom1 are delivered on the way to the nucleoplasm through the nuclear pores (Santos *et al*, 2018). In these processes, both ubiquitination and the interaction with adaptor protein syntenin-1 can regulate the intracellular trafficking of Prom1 (Karbanová *et al*, 2008; Yang *et al*, 2018). All these intriguing questions would request additional investigation particularly with regard to stem cells vs. transit amplifying cells.

Prom1 can play a role in the intercellular communication between stem cell populations and epithelial–mesenchymal interactions. In tissues whose development and homeostasis are involved in epithelial–mesenchymal interactions, in most cases, the mesenchyme plays some important roles in parallel to the epithelium. In the incisor CL region, Prom1 expression could be seen mainly in the epithelial transit amplifying cells and also in the opposing mesenchyme, where cells highly express signaling molecules such as Fgf3, Fgf7, and Fgf10 (Kettunen *et al*, 2000). We show that the epithelial Prom1 expression has a dominant role rather than the Prom1 in mesenchyme in controlling CL tissue development. The association of Prom1 with small extracellular vesicles needs a particular attention in this context. We previously demonstrated that Prom-1$^+$ vesicles are released from neural and hematopoietic stem and progenitor cells during the differentiation process, suggesting that loss of Prom1 is associated with stem cell maturation (Bauer *et al*, 2011). Upon release, Prom-1$^+$ vesicles can be endocytosed by stromal components as shown with bone marrow-derived stromal cells (Marzesco *et al*, 2005; Bauer *et al*, 2011). Therefore, the lack of Prom1 is our system might deregulate the intercellular exchange of biomaterials, i.e., biological information, and hence the potential feedback loop between stem cell populations and their cellular microenvironment. The selective release of Prom-1$^+$ vesicles from primary cilia of neural stem cells (Dubreuil *et al*, 2007) and the negative impact of K138Q mutation on the primary cilium structure (this study) would require further investigation to which our findings have brought cellular and molecular bases. Actually, it will be of interest to determine whether any biological molecular modification (e.g., ubiquitination) that would impede the Prom-1–Hdac6 interaction contributes to the cellular proliferation vs. differentiation. Future experiments are also needed to clarify the signals transduced from epithelium to mesenchyme direction without and with the losing Prom1 in the epithelium.

In a summary, our study indicates that Prom1 has important biological functions, particularly in stem cell activation and self-renewal, and provides direct evidence that the architecture of primary cilium affects stem cell maintenance and activation *in vivo* and demonstrates the signaling transduction mechanisms synchronized at cilia that are required for a proper stem cell response. These findings are likely to be relevant to other stem cell systems and ciliopathy conditions, and provide a mechanistic rationale that the strategy of targeting cancer stem cells through Prom1 can indeed direct cancer therapy and tissue regeneration (Pattabiraman & Weinberg, 2014).

# Materials and Methods

### Animals

All transgenic and WT animal breeding and procedures were approved by the institutional animal care and use committees at individual universities and in accordance with the guidelines and regulations for the care and use of laboratory animals in the corresponding countries: *Prom1*$^{-/-}$ (Zacchigna *et al*, 2009) at University of Lausanne, Switzerland and Max Planck Institute of Molecular Cell Biology and Genetics (MPI-CBG, Dresden), Germany; *Glis2*$^{LacZ/LacZ}$ (Kim *et al*, 2008) at the University of Iowa, USA; *Bmi1*$^{GFP/GFP}$ (Biehs

*et al*, 2013) at the University of California, San Francisco, USA; *IFT88*$^{flox/flox}$ (Haycraft *et al*, 2007) and *Rosa26*$^{CreER}$ (Badea *et al*, 2003) at the University of Oxford, UK, and CD1, C57BL/6 mice at the University of Plymouth, UK.

### Tamoxifen-mediated Cre recombination

At P21 and P22, 1 mg of tamoxifen (Sigma-Aldrich, T5648) was administered via I.P. injection to both *Rosa26*$^{CreER}$ x *IFT88*$^{flox/flox}$ or *IFT88*$^{flox/flox}$ mice. At P31, animals were sacrificed and heads fixed in neutral buffered formalin for 48 h. Samples were then transferred to 70% ethanol before immunohistochemistry preparation. For tamoxifen preparation, 50 μl of tamoxifen was administered at a concentration of 20 mg/ml. Tamoxifen was dissolved in 500 μl 100% ethanol and 4,500 μl of sunflower oil.

### CL tissue recombination assay

Cervical loops from P7 C57BL/6 and *Prom1*$^{-/-}$ lower incisors were microdissected under a Leica M80 stereomicroscope, then incubated in 1% Dispase II (Roche Diagnosis), and diluted in Ca$^{2+}$ and Mg$^{2+}$-free Hank's balanced salt solution (HBSS, Gibco) for 1 h. The samples were then transferred into Dulbecco's modified Eagle's medium (DMEM) containing 10% fetal bovine serum (Sigma-Aldrich) and 1% penicillin–streptomycin (Hyclone). The epithelium and mesenchyme were separated using 23G fine needles, then recombined as described in Fig 3E, and cultured on semisolid medium (Hu *et al*, 2006). The samples were harvested after 3 days in culture.

### Primary incisor tooth epithelial stem cell isolation and culture

Cervical loops were dissected under a Leica M80 stereomicroscope. Explants were incubated in 1% type I collagenase (Sigma-Aldrich) in HBSS with 1% penicillin–streptomycin and 1% Fungizone® Antimycotic (all from Thermo Fisher Scientific) for 60 min and dissociated by pipetting 20 times every 15 min. Collagenase was deactivated with equal amount of DMEM containing 10% fetal bovine serum (Sigma-Aldrich) and 1% penicillin–streptomycin. The cell suspension was then centrifuged at 1,000 × *g* for 5 min, and the resulting pellet was re-suspended in DMEM/F12 (Thermo Fisher Scientific) containing B27 supplements (Minus vitamin A version, Thermo Fisher Scientific), recombinant mouse EGF (20 ng/ml final concentration; carrier-free form, R&D Systems), recombinant mouse basic FGF (25 ng/ml final concentration; carrier-free form, R&D Systems), and 1% PS. From the second passage, cells started to express typical keratinocyte-like morphology. The exclusion of contamination of fibroblasts was confirmed by real-time RT–PCR on *Cytokeratin 14* and *Vimentin* (Appendix Fig S2E). All experiments were performed with cells between passages 4 and 10. For SHH stimulation experiments, cells were treated with indicated concentration of SHH protein (carrier-free form, R&D Systems) for 24 h before further analysis.

### Clonogenicity assay

A total of 2,000 single cells were seeded per well (12-well plate, Nunc) in triplicates and maintained in the indicated culture

conditions. Cells were fixed in 10% formalin (Sigma-Aldrich) for 30 min at room temperature (RT) and washed thrice with PBS. Colonies were visualized by crystal violet staining (Sigma-Aldrich) for 1 h at RT and then air-dried overnight before imaging. Stereo-scopic images were taken on Leica M80 through a Leica MC170 HD camera with an associated software (LAS, version: 4.4.0:454). The images were merged in Adobe Photoshop CS6, and number and size of colonies were quantified using Fiji 1.0 software (Schindelin *et al*, 2012).

### Organoid sphere culture

Passage 2 CLESCs and CLE mesenchymal cells were mixed at ratio of 1:1 and seeded at a density of 10,000 per well in 75 µl Matrigel on ice on a 96-well plate. After the gel became solidified at RT, a further 200 µl culture medium containing DMEM-F12 as basal medium and supplemented with 5% mouse B27 (Gibco), 5% mouse Amniomax (Gibco) 20 ng/ml mouse EGF, 25 ng/ml mouse bFGF, 0.1% Vitamin C (Sigma-Aldrich), and 2% Matrigel was topped up onto the gel. The culture medium was changed every 2 days.

### Plasmid construction and site-directed mutagenesis

Human *Prom1* expression plasmid was constructed by introducing the full-length coding sequence of human *Prom1* splice variant s2 (GenBank Accession number AF027208.1; Yang *et al*, 2008) into the enhanced yellow fluorescent protein (EYFP)-N1 vector (BD Bioscience Clontech). *Xho*I and *Not*I restriction sites were used for directional cloning. Note that *Xho*I/*Not*I digestion of pEYFP-N1 vector causes the excision of EYFP. The mutation of lysine 138 was introduced using QuikChange® II Site-Directed Mutagenesis Kit (Stratagene) according to the manufacturer's instructions. The oligonucleotides were 5′-CGTTGCTGTAAC**C**AATGTGGTGGAG-3′ [kt-5′$^{K→Q}$] and 5′-CTCCACC ACATT**G**GTTACA**G**CAACG-3′ [kt-3′$^{K→Q}$] as sense and antisense primers, respectively. The single nucleotide change from the original adenine to cytosine (underlined) leads to the encoding of a glutamine residue. In both cases, *Prom1* sequences within the expression plasmids were sequenced in order to confirm the absence of any reading mistakes introduced by the DNA polymerase and/or subcloning steps. Sequencing reactions were performed using Applied Biosystems 3730 Genetic Analyzer at the MPI-CBG facility (Dresden, Germany).

### MDCK cell culture and transfection

MDCK cells (strain II, American Type Culture Collection, #ATCC® CCL-34™) were cultured as described previously (Corbeil *et al*, 1999) and stably transfected with human Prom1 or Prom1 K138Q plasmids using nucleofection method with Amaxa® Nucleofector® Technology (Amaxa). Two days after transfection, cells were selected by introducing Geneticin (500 µg/ml, Gibco) into the culture medium and further expanded in a medium containing 250 µg/ml selective agent. To enrich the cell population express-ing recombinant Prom1 or K138Q mutant, cells were sorted using paramagnetic labeling with CD133 MicroBeads and LS columns (Miltenyi Biotec) following the manufacturer's instructions (Freund *et al*, 2006). Under these conditions, 80–95% of cells

expressed the transgene. To enhance its expression, 10 mM sodium butyrate was added to the culture medium for 16 h prior to immunocytochemistry experiments as described (Corbeil *et al*, 1999). To investigate the influence of Hdac6, cells were incubated for 16 h with 1 µM tubacin (SML0065, Sigma-Aldrich), 1 µM of the inactive analogue niltubacin (AbMole BioScience), or DMSO (1:10,000) as control.

### Transfection and tubacin treatment on CLESCs

Full-length mouse Prom1 and control vectors (Weigmann *et al*, 1997) were transfected into CLESCs using JetPEI (PolyPlus) according to the manufacturer's recommendations. Transfected CLESCs were grown under normal cell culture conditions for 48 h prior to analysis. For tubacin treatment, CLESCs were treated with 100 ng/ml SHH and with 10 µM of tubacin, 10 µM niltubacin (ab144537, Abcam), or DMSO for 24 h prior to proceeding with clonogenicity assay.

### Importazole treatment

Blocking of SHH-mediated nuclear translocation was achieved by treating cells simultaneously with SHH and 40 µM Importazole (SML0341, Sigma-Aldrich) or vehicle (DMSO) for the indicated time points in Fig 9B. CLE cells were then processed for further analysis as described in the "Proximity Ligation Assay (PLA)".

### Generation of Ki67p-Fucci lentiviral reporters and cell transfection

Open reading frames of the mAG-hGeminin(1/110)-pCSII-EF-MCS and mCherry-hCdt1(30/120)-pCSII-EF-MCS vectors (Sakaue-Sawano *et al*, 2008) kindly provided by Dr. Astsushi Miyawaki were PCR subcloned into pENTR-D-Topo plasmids (Life Technolo-gies) and sequence verified. Gateway recombination of the 1.5-kb Ki67 proximal promoter (Ki67p; Zambon, 2010) upstream of mAG and mCherry FUCCI reporters was conducted into 2k7 lentiviral vectors (Suter *et al*, 2006), enabling blasticidin and/or neomycin selection for cells expressing Ki67p-mCherry-hCdt1(30–120) and Ki67p-mAG-hGem(1–110), respectively. These Ki67p-Fucci constructs were transfected into the HEK293FT cell line, produc-ing lentiviral particles for cell transduction. CLESCs were incu-bated with the viral supernatant containing the plasmids and 10 µg/ml polybrene (Merck) for 2 h under normal culture condi-tions. After 2 h, the viral supernatant was replaced with normal growth medium. Infected CLE cells were selected using 10 µg/ml blasticidin (Sigma-Aldrich) and 250 µg/ml G418 (Geneticin Gibco).

### Flow cytometry

For flow cytometry, infected CLE cells were transfected with empty vector or mouse Prom1 (Weigmann *et al*, 1997) and then harvested after 72 h, fixed in 4% PFA for 30 min at RT, and analyzed using the BD FACSCanto II SOR (Beckman Coulter). Data were acquired using blue laser (488 nm) for mAzamiGreen signal and yellow-green laser (561 nm) for mCherry. Results were analyzed using the FlowJo software v10.3 (Tree Star Inc.).

### Laser capture microdissection

For each strain of mice, more than three frozen heads were cryosectioned at 10 μm thickness on a Leica CM1850 cryostat. Sections were mounted onto PEN Membrane Glass Slides (Thermo Fisher Scientific) and immediately stained using 1% Methyl Green (Sigma-Aldrich) in 0.1% diethyl pyrocarbonate (DEPC, Sigma-Aldrich)-treated water followed by four times for 30 s in DEPC water and then allowed to dry for 5 min. Laser capture microdissection was performed using CapSure® Macro LCM Caps (Thermo Fisher Scientific) on an ArcturusXT™ LCM instrument within 30 min of sectioning. Samples were stored in Tri-Reagent (Sigma-Aldrich) at −80°C until further processing.

### Real-time RT–PCR

Total RNAs were prepared using phenol–chloroform extraction methodology. Reverse transcription of total RNAs was performed using the High-Capacity cDNA synthesis kit (Thermo Fisher Scientific). Real-time RT–PCR analysis was performed using the SYBR Green I Master Mix (Roche) and primers listed in Appendix Table S1 on a LightCycler 480 Instrument II Real-Time PCR system (Roche Molecular Diagnostics) for 45 cycles. Murine *GapDH* gene was used as internal control for quantifying relative gene expression by using the $2^{-\Delta C_t}$ method. All analyses were performed with three replicates as described previously (Hu *et al*, 2012).

### Immunolabeling and confocal laser scanning microscopy

Frozen sections or CLESCs were mounted on Polysine slides (Thermo Fisher Scientific) or cultured on plastic bottom 96-well plates (Greiner Bio-One), respectively. Samples were fixed in 4% ice-cold paraformaldehyde (PFA, diluted in 10 mM PBS), for 30 min at RT, and then rinsed three times for 5 min in PBS-T (10 mM PBS containing 0.1% Triton X-100). Non-specific binding sites were first saturated by incubation in blocking buffer I (PBS-T containing 5% donkey serum, 0.25% cold water fish gelatine, and 0.25% bovine serum albumin) for 60 min. Primary antibodies (for details, see Appendix Table S1) were diluted in blocking buffer I and incubated overnight at 4°C. After three times 5-min washes in PBS-T, cells were incubated with appropriate fluorochrome-conjugated secondary antibodies (for details, see Appendix Table S1) diluted in blocking buffer I for 2 h at RT in the dark. Cells were counterstained with 4′,6-diamidino-2-phenylindole (1:10,000, DAPI, Sigma-Aldrich) for 10 min and then were mounted with DAKO Fluorescence Mounting Medium (DAKO).

For MDCK cells, they were cultured on fibronectin (BD Biosciences)-coated glass coverslips in silicon eight-well chambers (flexiPerm, Greiner Bio-One GmbH). Cells were rinsed with Ca/Mg buffer (phosphate-buffered saline containing 1 mM $CaCl_2$ and 0.5 mM $MgCl_2$) and incubated in blocking buffer II (Ca/Mg buffer containing 0.2% gelatine) for 30 min at 4°C. For cell surface staining, cells were incubated with mouse mAb CD133/1 (Miltenyi Biotec) diluted in blocking buffer II for 1 h at 4°C. After washing with PBS, they were fixed in 4% PFA in PBS for 30 min at RT. They were then rinsed and quenched for 10 min in PBS containing

50 mM $NH_4Cl$. Fixed cells were permeabilized and blocked with PBS containing 0.2% gelatine and 0.2% saponin (Panreac Applichem) for 30 min at RT. To visualize the primary cilia, cells were incubated with mouse mAb anti-AcTub (Sigma-Aldrich) and/or rabbit polyclonal (p)Ab Arl13b (Proteintech) for 30 min in permeabilization buffer. In some experiments, cells were incubated with rabbit pAb anti-Hdac6 (Thermo Fisher Scientific). In all cases, the appropriate Alexa 488/546/647-conjugated goat anti-mouse IgG$_1$ (Molecular Probes, Life Technologies) and goat anti-mouse IgG$_{2b}$ antibodies (Thermo Fisher Scientific) were used to detect the anti-Prom1 or anti-AcTub antibody, respectively. Alexa 546-conjugated goat anti-rabbit IgG antibody (Molecular Probes, Life Technologies) was used to detect the anti-Arl13b or anti-Hdac6 antibody. Nuclei were counterstained with DAPI (1 μg/ml, Sigma-Aldrich). Samples were washed three times in PBS containing 0.2% gelatine, two times in PBS. The samples were scanned with either a Zeiss LSM 700 or a Leica SP5 confocal microscopes.

### Terminal deoxynucleotidyl transferase (TdT) dUTP Nick-End Labeling (TUNEL) assay

For TUNEL assays, apoptosis of CLESCs or on frozen sections was assessed using the *In situ* Cell Death Kit, Fluorescein (Roche), according to the manufacturer's instructions. Briefly, samples were fixed with ice-cold 4% PFA at RT for 20 min followed by a 30-min wash with PBS and 2-min permeabilization step using 0.1% Triton X-100 in PBS. Samples were then incubated with freshly prepared enzyme and label solution for 1 h at 37°C under humid conditions and mounted using DAKO Fluorescence Mounting Medium (DAKO). Negative control samples were incubated in the absence of enzyme solution. Positive control samples were treated with TURBO DNase (Ambion) for 10 min at RT prior to adding enzyme and label solution. Apoptotic bodies were excited at 450–500 nm and detected at 515–565 nm using a Leica SP5 confocal microscope.

### Three-dimensional (3D) reconstruction

Micrographs in Fig 6I and J were prepared from data files using Velocity 3D Image Analysis Software (PerkinElmer). Primary cilia of MDCK cells were quantified using Huygens Professional Software (Scientific Volume Imaging, Hilversum, the Netherlands). After deconvolution of Z-stack images, the advanced object analyzer was used to obtain their length and mean fluorescence intensity (MFI).

For the other figures, fluorescence z-stack images acquired from confocal microscopy were used to reconstruct primary cilia in 3D using Imaris 9.0 (Bitplane). Measurements of primary cilia dimensions were performed manually using the Measurement tools in Imaris 9. When necessary, 3D reconstructed primary cilia images were rotated to render a cross-section appropriate for measurements.

For incisor tooth cervical loop volume analysis, the structure was outlined based on basement membrane from approximately 20 consecutive 20-μm-thick coronal sections of the apical end of the mouse incisor (*n* = 3 animals per genotype). BioVis software (version 3.1.1.11; Biovis3d) was used to render 3D reconstruction of the Sox2-positive region of the cervical loop and to calculate the volume.

## Phase-contrast microscopy

Dome formation was observed by phase-contrast microscopy using a high-resolution digital camera (Canon PowerShot A620) attached to a Zeiss Axiovert 40C microscope. Micrographs were processed using Adobe Photoshop CS6.

## Scanning electron microscopy (SEM)

For MDCK cells, samples for SEM analysis were prepared as described (Florek *et al*, 2007). Briefly, cells grown for 6 days post-confluence (dpc) on fibronectin-coated coverslips were fixed in 2% glutaraldehyde overnight at 4°C. After dehydration in an acetone gradient (25–100%), cells were critical-point-dried in a $CO_2$ system (Critical Point Dryer CPD 300, Leica). Samples were mounted on aluminum stubs, coated with gold in a sputter coater (SCD 050, BAL-TEC GmbH), and examined with a scanning electron microscope (JSM-7500F, JEOL).

For analyzing tooth enamel structures, isolated incisor teeth were fixed in 3% glutaraldehyde (Sigma-Aldrich) in PBS for 4 h and then incubated in ethanol for 5 min each at concentration of 50%, 70%, 90%, 95%, and 3 times for 30 min at concentration of 100%. Samples were then dried for 24 h, gold-sputtered, and observed under a XL20 scanning microscope (Philips).

## Differential centrifugation

Prom1-transfected MDCK cells were grown in six-well dishes. After gently washing them with PBS, they were supplied with fresh culture medium (3 ml) and incubated for 16 h. Afterward, conditioned media were collected, supplemented with Complete™ protease inhibitor cocktail (Roche Diagnostics GmbH, Manheim, Germany) and subjected to differential centrifugation as follows: 5 min at $300 \times g$ (P1); 20 min at $1{,}200 \times g$ (P2); 30 min at $10{,}000 \times g$ (P3); and 1 h at $200{,}000 \times g$ (P4). Pellets were re-suspended in Laemmli buffer. In parallel, adherent cells were harvested and centrifuged for 5 min at $300 \times g$. The resulting pellets were lysed in solubilization buffer I (1% NP-40, 0.5% deoxycholate, 0.1% sodium dodecyl sulfate, 150 mM NaCl, 50 mM Tris–HCl, pH 8.0) supplemented with Complete protease inhibitor for 30 min at 4°C. Detergent extracts were centrifuged at 4°C (10 min, $16{,}000 \times g$), and resulting supernatants were mixed with Laemmli buffer. All samples were analyzed by immunoblotting for Prom1, β-actin, and acetylated α-tubulin (see below). For quantification, we combined Prom1 immunoreactivity associated with cell extract, P1 and P2 fractions, and designated them as *Cell fraction,* while materials found in P3 and P4 were referred to as *Vesicle fraction.*

## Immunoisolation

Prom1—MDCK cells were grown in a T75 flask, and at 6 dpc, 10 mM sodium butyrate was added for 16 h. Afterward, cells were solubilized for 30 min on ice in solubilization buffer II (0.2% Triton X-100, 150 mM NaCl, 20 mM Tris–HCl, pH 7.5) containing Complete™ protease inhibitor cocktail (Roche Diagnostics GmbH), 2 mM sodium orthovanadate, 10 mM sodium fluoride (both from New England Biolabs Inc., Ipswich, MA) and 10 mM β-glycerophosphate (Sigma-Aldrich). After centrifugation (10 min, $16{,}000 \times g$, 4°C) to remove insoluble materials, the detergent lysate (1.6 ml) was incubated with 130 μl of magnetic basic MicroBeads Miltenyi Biotec) for 30 min, and using the magnetic separation with MS column (Miltenyi Biotec), the proteins that non-specifically bind to magnetic beads were removed. Pre-cleared lysates were then divided in two fractions that were further incubated with either 65 μl of anti-CD133 magnetic MicroBeads or 130 μl anti-FITC magnetic MicroBeads (Miltenyi Biotec) as negative control, for 1.5 h at 4°C prior to magnetic separation using MS columns. Materials retained in columns were washed 6 times with 1 ml of ice-cold PBS containing 2 mM EDTA and 0.1% Triton X-100, and then eluted with pre-heated (95°C) Laemmli sample buffer according to the manufacturer's instructions. Flow-through materials were collected, subjected to methanol/chloroform precipitation, and re-suspended in Laemmli sample buffer. Bound and flow-through materials were analyzed by immunoblotting for Prom1 and Arl13b (see below) and their ratio quantified.

Arl13b—Cell lysates of MDCK cells were acquired as described above, and an aliquot was removed as input sample. Pre-cleared lysates were incubated first with 3.5 μl of anti-Arl13b pAb for 30 min at 4°C, and then, 100 μl μMACS Protein A MicroBeads was added and incubated for another 1 h at 4°C. Bound fraction was further magnetically separated and processed, together with its flow-through material, as described above.

## Immunoblotting

For MDCK cells, proteins were separated by sodium dodecyl sulfate–polyacrylamide gel electrophoresis (SDS–PAGE; 7.5% or 8%) and transferred to polyvinylidene fluoride membranes (pore size 0.45 μm, Merck) using a semi-dry transfer cell system (Cti). After transfer, membranes were incubated overnight at 4°C in blocking buffer III (PBS containing 0.3% Tween-20 and 5% milk powder) prior to 1-h incubation at RT with either primary antibody against Prom1 (mouse mAb anti-CD133, clone 80B258 (Karbanová *et al*, 2008) or clone W6B2C1 (Miltenyi Biotec GmbH)) or anti-Arl13b pAb diluted in blocking buffer III. As loading controls, mouse mAb AC-74 against β-actin (Sigma-Aldrich) and mouse mAb anti-acetylated tubulin were used. In all cases, antigen–antibody complexes were detected using appropriate horseradish peroxidase-conjugated secondary antibodies (Jackson Immuno-Research Inc.) and visualized with enhanced chemiluminescence reagents (ECL system; GE Healthcare Life Sciences). Membranes were exposed to Amersham Hyperfilm ECL (GE Healthcare Life Sciences). Fiji was used for the quantification of immunoblotted materials.

For CLESCs, $10 \times 10^6$ CLE cells were washed with ice-cold HBSS and then collected. Cells were spun down at 8,300 g for 10 min. Supernatant was removed, and pellet was used to extract nuclear and cytoplasmic protein fraction according to NE-PER Nuclear and Cytoplasmic Protein Extraction (Thermo Fisher Scientific). Fractions were quantified using BCA Protein Assay (Thermo Fisher Scientific). Protein separation and membrane transfer were performed using NuPage precast gels, MOPS, and transfer buffers (Thermo Fisher Scientific). Primary and secondary antibody incubations and washing steps were performed simultaneously using the iBind system (Thermo Fisher Scientific). Detection of bands was done using C-Digit blot scanner (LI-COR).

## Proximity ligation assay

For *in situ* PLA, sagittal mouse incisor tooth sections were prepared and permeabilized as for immunofluorescence labeling. Primary antibodies targeted against Prom1 (ORB129549, Biorbyt) and Glis2 (Li *et al*, 2011) were conjugated with MINUS (Sigma-Aldrich) and PLUS (Sigma-Aldrich) probes, respectively. PLA experiments were then performed with the Duolink kit (Sigma-Aldrich) according to the manufacturer's instructions. Briefly, sections were blocked for 1 h at RT with 5% donkey serum and 0.25% cold water fish gelatine in PBS-T, incubated with conjugated primary antibodies Prom1-MINUS and GLIS2-PLUS overnight at 4°C. Ligation of the bound probes and subsequent amplification of the signal was performed exactly as recommended in the manufacturer's protocol. The negative control experiments were performed in parallel and one of the conjugated primary antibodies was omitted (anti-Prom1 antibody conjugated with negative probes).

## Chromatin immunoprecipitation

Potential Glis2 transcription factor binding sites in the STAT promoter were determined using Genomatix software. Primers were designed using Primer3 and validated by RT–qPCR. ChIP assays were performed using ChIP-IT Express Kit (Active Motif) as per the manufacturer's recommendations. In brief, after crosslinking, nuclei were purified and chromatin was sheared by sonication using Bioruptor Pico sonicator (Diagenode) at frequency of five times, 15 s ON/30 s OFF to achieve enriched fragments at size of 500–700 bp. Sheared chromatin was incubated overnight with antibody directed against GLIS2 (ARP30037_P050, AVIVA). Matching rabbit IgG Isotype antibodies were used as a negative control for the immunoprecipitation. Immunoprecipitated chromatin was then incubated with protein G magnetic beads, washed, and eluted. After reversal of the crosslinks and purification of DNA, precipitated DNA was analyzed by RT–qPCR.

## Small interfering RNAs

Small interfering RNAs (siRNAs) were commercially synthesized (Sigma-Aldrich) using a 486-bp-long cDNA targeting *Prom1* (EMU055321) targeting mouse Prom1 sequence NM_008935 or cDNA targeting *RLUC* gene (EHURLUC) as a transfection control. Knockdown of Prom1 was achieved by transfecting CLESCs with 20 nM siRNA (final concentration) targeting Prom1 by using INTERFERin (PolyPlus) according to the manufacturer's recommendations. The RLUC control was transfected in the same manner as esiProm1. siHdac6 and siArl13b were performed using the same condition (for reagent details please see Appendix Table S1). Cells were harvested at 48 h after transfection for further analysis.

## Short hairpin RNA (shRNA) (Generation of lentivirus and shRNA delivery)

Co-transfection of pLKO.1 (control), pLKO.1-shGlis2 (clones 5 and 7) (Li *et al*, 2011), and pLKO.1-shProm1 (for details, see Appendix Table S1) with packaging vectors was done on 293FT cells, and viral supernatant was collected after 48 h and 72 h.

Supernatant was then filtered through a 0.45-μm filter (Merck) to remove cell debris. CLE was then incubated for 2 h with the collected supernatant in the presence of polybrene (8 μg/ml, Merck) at normal culture conditions. After 2 h, viral supernatant was replaced with normal culture medium. The next day, CLE was grown indefinitely in normal medium supplemented with puromycin (10 μg/ml, Sigma-Aldrich).

## X-Ray microscopy-based microCT analysis

Samples were imaged at the Small Animal Imaging Core of the University of Iowa using the Zeiss Xradia 520 Versa (Pleasanton, CA). The 3D X-ray source acquisition settings were 120 kV, 10 W, and 0.4× magnification. Source distance from the sample was 27 mm, and the detector distance was 170 mm. A total of 1,601 projections with a 1-s exposure time over 360 degrees of rotation were acquired. Using the Reconstructor Scout-and-Scan software (Zeiss, Pleasanton, CA), projections were reconstructed with downsampling of 1, with beam hardening correction to yield a final pixel size of 25 μm. The generated DCOM files were reconstructed into 3D images and analyzed in Imaris software (9.0, Bitplane).

## Statistical analysis

Statistical analyses were performed using GraphPad Prism version 6.00 for Mac (GraphPad Software). Two-tailed Fisher's exact test was used for the analysis of Prom1$^+$ cells without cilium. The length and MFI of ciliary tubulin were analyzed using the Mann–Whitney test. For the box–whisker plot, the boxes represent data from 25th to 75th percentiles and 100% within the whiskers. Horizontal lines within the box represent median values. Two-tailed unpaired Student's *t*-test was applied for the evaluation of dome formation, nuclei numbers regarding dome formation or proliferation, Prom1 immunoreactivity associated with membrane vesicles, and the ratio of ciliary Hdac6/AcTub and of Arl13b/Prom1. Observed differences were regarded as significant if the calculated *P*-values were ≤ 0.05. *$P < 0.05$; **$P < 0.01$; ***$P < 0.001$.

All quantification and real-time RT–PCR results were presented using style of mean and standard deviation (error bars).

**Expanded View** for this article is available online.

## Acknowledgements

We would like to thank group members of the Corbeil and Hu laboratories for critical discussions of the works, and ZLS. Brookes, L. Belfield, and J. Davies for critical reading. We thank G.P. Dotto for initial support of the project; T. Edwards for Hu's Lab's management assistance; S. Walsh and M.R. Acevedo for assistance on X-ray microscopy scanning; M.C. Reymond for assisting SEM; T.P. Yao for Hdac6 antibodies; and A. Miyawaki for mAG-hGeminin (1/110)-pCSII-EF-MCS and mCherry-hCdt1(30/120)-pCSII-EF-MCS vectors. This work was supported by the National Institutes of Health to M.A. (R01DK090326-01A1) and to O.D.K (R35-DE026602), the National Natural Science Foundation of China to H.Z. (30801289 and 81371138), the KTRR Prize studentship for C.R.C, the Arthritis Research United Kingdom (ARUK) Centre for Osteoarthritis Pathogenesis Grant to A.W. (20205 and 21621), the VIB TechWatch program, long-term structural Methusalem funding of the Flemish Government, grants from

the Research Foundation Flanders (FWO-Vlaanderen), Foundation against Cancer (2016-078), Kom op Tegen Kanker (Stand up to Cancer, Flemish Cancer Society) and ERC Advanced Research Grant (EU-ERC743074) P.C., the Deutsche Forschungsgemeinschaft (DFG, SFB 655 A2, and B3) to W.B.H. and D.C., the European Union Marie Skłodowska-Curie actions (618930, OralStem FP7-PEOPLE-2013-CIG), the European Regional Development Fund, and the Biotechnology and Biological Sciences Research Council of the UK (BB/L02392X/1) to B.H.

## Author contributions

DS, KT, HZ, JK, YG, JVW, PRCG, TG, and BH performed experiments, and collected and analyzed data. HJ, DL, AMJ, and MA prepared *Glis2* knockout mice and related reagents. XW, PM, and ODK prepared *Bmi1* knockout mice. CRC and AW prepared *Rosa*$^{ERT2}$ and *IFT88*$^{flox/flox}$ mice. PC prepared *Prom1* knockout mice. SA performed tooth surface SEM and analyzed data. ACZ designed and created the *Ki67-FUCCI* system. KJL and SL assisted experimental design. CT provided guidance throughout the works. DC, WBH, and BH conceived and designed most of the experiments reported, analyzed data and wrote the manuscript.

## Conflict of interest

The authors declare that they have no conflict of interest.

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
