## [Review Process File · The EMBO Journal]

Prominin-1 controls stem cell activation by orchestrating ciliary dynamics

Donald Singer, Kristina Thamm, Heng Zhuang, Jana Karbanová, Yan Gao, Jemma Victoria Walker, Heng Jin, Xiangnan Wu, Clarissa R. Coveney, Pauline Marangoni, Dongmei Lu, Portia Rebecca Clare Grayson, Tulay Gulsen, Karen J. Liu, Stefano Ardu, Angus Wann, Shouqing Luo, Alexander C. Zambon, Anton M. Jetten, Christopher Tredwin, Ophir D. Klein, Massimo Attanasio, Peter Carmeliet, Wieland B. Huttner, Denis Corbeil & Bing Hu

Review timeline:	Submission date:	24th May 2018
	Editorial Decision:	29th Jun 2018
	Revision received:	12th Sep 2018
	Editorial Decision:	18th Oct 2018
	Revision received:	22nd Oct 2018
	Accepted:	23rd Oct 2018

Editor: Daniel Klimmeck

Transaction Report:

(Note: Please note that the manuscript was previously reviewed at another journal and the reports were taken into account in the decision making process at The EMBO Journal. Since the original reviews are not subject to EMBO's transparent review process policy, the reports and author response cannot be published here. With the exception of the correction of typographical or spelling errors that could be a source of ambiguity, letters and reports are not edited. The original formatting of letters and referee reports may not be reflected in this compilation.)

1st Editorial Decision	29th Jun 2018
---------------

Thank you again for your interest and the submission of your manuscript (EMBOJ-2018-99845) to The EMBO Journal. My apologies for getting back to you with delay due to delayed input from the arbitrating expert as well as detailed discussions in the team. We have carefully assessed your manuscript and the point-by-point response provided to the referee concerns that were raised during review at a different journal. In addition, and as mentioned before, we decided to involve an arbitrating expert to evaluate the revised version of your work. This advisor has been instructed to focus solely on the technical robustness and conclusiveness of the new data added as well as on the suitability of your work for publication in The EMBO Journal.

As you will see from the report provided below, the arbitrating advisor states the high interest and is in light of your additional input supportive of further consideration at The EMBO Journal.

Based on the support provided by the advisor and our own assessment, we realise that you would be potentially able to address the issues raised by the referees in a revised version of the manuscript along the lines of information provided in the point-by-point response.

I judge the comments of the referees to be generally reasonable and can - based on your sensible

preliminary response - offer to invite you to revise your manuscript experimentally to address the referees' concerns. I agree that in particular the aspect of functional causalities between Prom1 and Arl13b-HDAC6 - ciliary dynamics or alternatively recapitulating the tooth development phenotype in an independent way by changing cilia number or size would need to be conclusively addressed in a revised version of the manuscript to move towards publication.

Please submit a revised version of the manuscript using the link enclosed below, addressing the reviewers' comments.

ARBITRATING ADVISOR'S COMMENTS:

'In general, I agree that the performance of the suggested experiments would go a considerable way to addressing the reviewers' critiques; I think the reviewers did a good job, and agree showing causality and anchoring the idea that the Hedgehog pathway is affected is the key to this being a really impactful study.'

1st Revision – authors' response

12th Sep 2018

Arbitrating advisor

Arbitrating advisor's comments:

'In general, I agree that the performance of the suggested experiments would go a considerable way to addressing the reviewers' critiques; I think the reviewers did a good job, and agree showing causality and anchoring the idea that the Hedgehog pathway is affected is the key to this being a really impactful study.'

Answer from the authors:

We thank for the advisor's comments and now have intensively revised our manuscript based on the advisor and reviewers' comments. We now can prove that indeed the Hedgehog pathway is affected, particularly through Glis2. In addition, we have introduced the other two transgenic lines to delete IFT88, the key intrinsic cilia component and showed the impact on incisor epithelial stem cells are significant that further proved our findings. For details please see our below answers to the reviewers and the revised manuscript.

2nd editorial decision

18th Oct 2018

Thank you for submitting your revised manuscript for consideration by The EMBO Journal and your patience with our response. Your revised study was sent back to the arbitrating advisor for re-evaluation, and we have received his/her comments, which I enclose below. As you will see the expert finds that the major concerns have been sufficiently addressed and is in favour of publication, pending a number of minor remaining points are conclusively considered.

Thus, we are pleased to inform you that your manuscript has been accepted in principle for publication in The EMBO Journal, pending the minor issues regarding data interpretation and annotation, as well as manuscript formatting, as outlined below, which need to be adjusted at re-submission.

ARBITRATING ADVISOR'S COMMENTS:

In this extensively revised study, Singer et al. investigated the role of Prominin-1 (Prom1)/CD133 and the primary cilia in the process of the stem cells (SCs) activation. As a model for stem cell progression, the authors, for the first time, used the mouse incisor tooth, which represents a quite interesting and useful tool for studying the activation and progression of the SCs into transit amplifying cells (TACs). Using this model, Singer et al. showed that the cholesterol-binding glycoprotein Prom1 plays a role in SC activation by regulating ciliary dynamics through an interaction with Arl13b and Hdac6, in which Prom1-Arl13b interactions are associated with SCs and Prom1-Hdac6 with TACs. It is known that the fate of SCs is highly dependent on the proper coordination of multiple signaling pathways, including cilia-associated Hedgehog signaling (HHS). Thus, the authors report that the Prom1 control of ciliary dynamics also affects the response of SCs to the HHS. One of the most interesting observations from the current study is of the translocation of Prom1-Arl13b into the nucleus, although the mechanisms and the role of this translocation were not completely investigated. Additionally, the authors also suggested that Prom1 could regulate Hedgehog signaling in the SCs through the interaction with the transcription factor Glis2 and activation of its downstream target Stat3.

In general, the authors very significantly improved the present study by addressing reviewers' comments, most

importantly, by showing the direct interaction between Arl13b and Prom1. Overall, the conclusions are made based on high quality data.

There are some minor points that would be good to address:

1. Figure 5C. The authors say that Prom1 overexpression leads to an "increased proportion of cells at the G0 and G2/M phases". Could the authors explain why there is a simultaneous increase of both of these cell fractions?
2. Figure 5 D and E. The authors say: "only in the presence of SHH, we found the cell ciliary dynamics were correctly associated with cell cycle". Could you please explain and provide images describing the "incorrect" association of the ciliary dynamics with the cell cycle (in the absence of SHH)?
3. Figure 7 F and G. Could the authors explain why there is an increase of secretion of Prom1+ vesicles from the Prom1-mutated cells?

2nd Revision - authors' response

22nd Oct 2018

Reviewer's comment:

There are some minor points that would be good to address:

1. *Figure 5C. The authors say that Prom1 overexpression leads to an "increased proportion of cells at the G0 and G2/M phases". Could the authors explain why there is a simultaneous increase of both of these cell fractions?*

Answer from the authors:

The findings indicate that Prom1 is important for stem cell maintenance and proliferation, which fit with the function of SHH in parallel. We have also inserted a sentence to explain it at the beginning of page 12: "...and Prom1 overexpression in the CLESCs significantly increased the proportion of cells at G0 and G2/M phases (Fig 5C and Appendix Fig 5A), similar to the effect of SHH treatment (Appendix Fig 5B), suggesting that Prom1 was indeed critical for stem cell self-renewal and proliferation."

Reviewer's comment:

2. *Figure 5 D and E. The authors say: "only in the presence of SHH, we found the cell ciliary dynamics were correctly associated with cell cycle". Could you please explain and provide images describing the "incorrect" association of the ciliary dynamics with the cell cycle (in the absence of SHH)?*

Answer from the authors:

We have now included a new panel (D) in Figure 5, where we show clearly that only when SHH was added into the medium the Ki67-FUCCI CLESCs produced primary cilia.

Reviewer's comment:

3. *Figure 7 F and G. Could the authors explain why there is an increase of secretion of Prom1+ vesicles from the Prom1-mutated cells?*

Answer from the authors:

The fact that Hdac6 does not interact with K138Q Prom1 leads to the deacetylation of tubulin, and hence the cilium disassembly. Given that the microtubules are essential to support the structure of the cilium and the strong preference of Prom1 for membrane with a high curvature, membrane vesicles are formed along the cilium. The "pearling" morphology of remaining long cilia in cells expressing K138Q mutant illustrated well the phenomenon. We have previously demonstrated that membrane cholesterol, i.e. an interacting partner of Prom1, is also involved in such mechanism (Marzesco et al., FEBS Lett. 2009).

We added one sentence in the revised manuscript to explain it:

“Thus, the reduction of AcTub observed in primary cilia of K138Q mutant-expressing cells might result in the fission of Prom1⁺ vesicles concomitant with the reduction of ciliary length.”

Corresponding Author Name: Dr Denis Corbeil and Dr Bing Hu

Journal Submitted to: the EMBO Journal

Manuscript Number: EMBOJ-2018-99845R